# Mosquito salivary apyrase regulates blood meal hemostasis and facilitates malaria parasite transmission

Zarna Rajeshkumar Pala[1], Thiago Luiz Alves e Silva[1], Mahnaz Minai[2], Benjamin Crews[3], Eduardo Patino-Martinez[4], Carmelo Carmona-Rivera[4], Paola Carolina Valenzuela Leon[1], Ines Martin-Martin[1,8], Yevel Flores-Garcia[5], Raul E. Cachau[6], Liya Muslinkina[7], Apostolos G. Gittis[7], Naman Srivastava[1], David N. Garboczi[7], Derron A. Alves[2], Mariana J. Kaplan[4], Elizabeth Fischer[3], Eric Calvo[1] & Joel Vega-Rodriguez[1] ✉

The evolution of hematophagy involves a series of adaptations that allow blood-feeding insects to access and consume blood efficiently while managing and circumventing the host's hemostatic and immune responses. Mosquito, and other insects, utilize salivary proteins to regulate these responses at the bite site during and after blood feeding. We investigated the function of *Anopheles gambiae* salivary apyrase (AgApyrase) in regulating hemostasis in the mosquito blood meal and in *Plasmodium* transmission. Our results demonstrate that salivary apyrase, a known inhibitor of platelet aggregation, interacts with and activates tissue plasminogen activator, facilitating the conversion of plasminogen to plasmin, a human protease that degrades fibrin and facilitates *Plasmodium* transmission. We show that mosquitoes ingest a substantial amount of apyrase during blood feeding, which reduces coagulation in the blood meal by enhancing fibrin degradation and inhibiting platelet aggregation. AgApyrase significantly enhanced *Plasmodium* infection in the mosquito midgut, whereas AgApyrase immunization inhibited *Plasmodium* mosquito infection and sporozoite transmission. This study highlights a pivotal role for mosquito salivary apyrase for regulation of hemostasis in the mosquito blood meal and for *Plasmodium* transmission to mosquitoes and to the mammalian host, underscoring the potential for strategies to prevent malaria transmission.

Malaria, a debilitating and sometimes fatal infectious disease, continues to pose a major public health challenge, with ~249 million cases and 608,000 deaths globally in 2022[1]. The transmission of the causative agent, *Plasmodium* parasite, occurs via the bite of an infected *Anopheles* mosquito. The parasite undergoes sexual reproduction and differentiates into male and female gametes inside the mosquito midgut after ingesting a blood meal from an infected human, producing motile ookinetes that invade the midgut epithelium to form oocysts that produce the infectious sporozoites[2]. The development of the parasite in the mosquito midgut and transmission of sporozoites to humans represent critical bottlenecks in the life cycle, making them attractive targets for the development of malaria interventions. Recent

---

studies have shown that *Plasmodium* co-opts host proteins, such as the serine protease plasmin, to facilitate infection in both the mosquito vector and the human host[3–5].

Fibrinogen and fibrin are vital for maintaining hemostasis and are pivotal players in processes such as thrombosis, wound healing, and numerous other biological functions as well as pathological conditions. Fibrinogen, initially in a soluble form, transforms into an insoluble clot or gel when converted to fibrin. This conversion is catalyzed by the serine protease thrombin, which is activated through a cascade of enzymatic reactions triggered by factors released by vessel wall injury, activated blood cells, or the presence of a foreign surface. The formation of a mechanically stable clot is imperative to prevent blood loss, a phenomenon referred to as hemostasis. Fibrin clots are subsequently degraded by the fibrinolytic system in a process called fibrinolysis, operating through a series of enzymatic reactions with both positive and negative feedback mechanisms. A delicate equilibrium between clotting (the conversion of fibrinogen to fibrin), and fibrinolysis (the proteolytic disintegration of the clot) is maintained in vivo (Fig. 1a). Beyond its role in fibrin clot formation, fibrinogen is also essential in an earlier stage of hemostasis known as "primary hemostasis." This involves platelet aggregation, ultimately leading to the formation of a platelet "plug" at the site of vascular injury. Following the completion of its hemostatic function, clots formed in vivo are typically dissolved by the fibrinolytic system, thereby restoring normal blood flow[6].

Plasmin, a serine protease derived from its inactive precursor, plasminogen, is the principal enzyme involved in fibrinolysis. Notably, plasmin also cleaves extracellular matrix (ECM) proteins and activates certain proteases and growth factors. Fibrinolysis commences with the conversion of plasminogen to plasmin, primarily occurring on the fibrin surface, and is followed by the degradation of fibrin by plasmin. Plasminogen activators, such as tissue-type plasminogen activator (tPA) and urokinase-type plasminogen activator (uPA), are responsible for initiating the conversion of plasminogen to plasmin. In humans, the activity of the fibrinolytic system is mainly regulated by the serine protease inhibitors plasminogen activator inhibitor-1 (PAI-1), a potent inhibitor of tPA and uPA, and α2-antiplasmin, which directly inhibits plasmin. In a positive feedback loop, the basal activity of tPA and uPA activates plasminogen into plasmin, and plasmin further activates tPA and uPA thereby, enhancing the overall plasminogen activation[7]. We recently demonstrated that *Plasmodium* gametes hijack tPA to activate plasminogen on their surface, allowing it to degrade fibrin in the blood bolus and facilitate gamete migration[3]. Moreover, we showed that sporozoites also hijack plasmin for degradation of extracellular matrix (ECM) proteins, which facilitates its motility in the skin and liver[3].

Mosquito saliva contains numerous proteins with anti-hemostatic and immunomodulatory functions that not only play a role in facilitating blood feeding, but also impact pathogen transmission[8–12]. While the role of mosquito saliva and some salivary proteins in *Plasmodium* sporozoite transmission has been studied, our understanding of their impact on *Plasmodium* transmission from humans to mosquitoes, and from mosquitoes to humans, remains limited. Recent studies showed that the salivary proteins saglin and salivary gland surface protein 1 (SGS1), ingested during blood feeding, impact *Plasmodium* midgut infection however, although the mechanism of action is still unknown[13,14]. In this study, we provide the mechanism by which mosquito salivary apyrase, ingested during blood feeding, enhances fibrinolysis (degradation of fibrin polymers) and inhibits platelet activation in the mosquito blood bolus, which facilitates *Plasmodium* infection in the mosquito vector and thus, transmission to the mammalian host.

## Results
### Mosquito saliva activates human tissue plasminogen activator (tPA)
To evaluate the impact of mosquito salivary proteins on the fibrinolytic system, we investigated the ability of *Anopheles gambiae* salivary gland protein extracts and saliva to activate single-chain tissue plasminogen activator (sc-tPA). Using a tPA fluorogenic substrate (D-Val-Leu-Lys-7-amido-4-methylcoumarin), we observed increased tPA activity in response to both salivary gland extracts or saliva (Fig. 1b–d). Activation of tPA was lost upon heating the salivary glands to 65 or 100 °C (Fig. 1c), suggesting a salivary protein as the tPA activator. To identify the putative salivary tPA activator, we fractionated salivary proteins by size-exclusion chromatography and identified fraction Z8 as the strongest tPA activator, whereas the adjacent fractions Z7 and Z9 activated tPA at lower levels (Fig. 1e). Mass spectrometry analysis of these fractions identified a total of 152 unique proteins (Supplementary Data 1), which were shortlisted to eight potential tPA activators (Supplementary Data 2 and Fig. 1f) based on the presence of secretion signal, presence in adjacent fractions, and absence from fractions A5 and B3 which did not activate tPA (Supplementary Data 1). Fractions A5 and B3 were randomly selected for mass spectrometry analysis as representative of fractions not showing tPA activity. The eight candidates were expressed in HEK293 cells and purified (Fig. S1a), and only AGAP011026-PA, annotated as a 5' nucleotidase ecto (Ag5'NTE) in VectorBase[15], activated tPA in the fluorogenic assay at levels comparable to saliva (Fig. 1g and S1b), identifying salivary *An. gambiae* 5' NTE as a tPA activator. Next, using a colorimetric assay[3] we showed that rAg5'NTE-activated tPA resulted in higher activation of plasminogen to plasmin when compared to the positive controls of plasmin, or plasminogen activation by sc-tPA (Fig. S1c). These results demonstrate the role of Ag5'NTE in the activation of tPA. Based on our previous report showing that the fibrinolytic system enhances parasite infection in the mosquito and transmission to the mammalian host[3], we hypothesized that Ag5'NTE may facilitate *Plasmodium* transmission by enhancing the activation of fibrinolysis.

### Mosquito salivary apyrase is the tPA activator
The 5' nucleotidase ecto is an extracellular protein, normally attached to the membrane by a glycosyl phosphoinositol (GPI) anchor or secreted, that hydrolyzes AMP to adenosine and phosphate[16]. However, the AGAP011026-PA sequence does not have the predictive site for a GPI anchor indicating that it is secreted[17,18]. Using a biochemical colorimetric assay to measure phosphate release as an indicator of AMP hydrolysis[19], we found that rAg5'NTE did not show any activity, whereas a strong activity was observed in the positive control with human 5' NTE (CD73) (Fig. 2a). 5' NTE is a member of the 5'-nucleotidase family that also include apyrases, which hydrolyze ATP and ADP to AMP in the presence of a bivalent metal ion[20,21]. Upon incubation of rAg5'NTE with either ATP or ADP in the presence of calcium, we observed significant release of inorganic phosphate at levels similar to the positive control potato apyrase (Fig. 2b, c). Apyrases belong to three families: the 5' nucleotidase family which includes apyrases from *Aedes aegypti*[22], *An. gambiae*[23] and *Triatoma infestans*[24]; a novel apyrase family which includes apyrases from *Cimex lectularius*[25], *Phlebotomus papatasi*[26], *Lutziomyia longipalpus*[27], *Drosophila melanogaster*[27], and humans[28], and the CD39 family, which includes apyrases of humans[29], and possibly, fleas[30]. The 5' nucleotidase type functions in the presence of $Ca^{2+}$ or $Mg^{2+}$, and the *Cimex* type works only with $Ca^{2+}$[31]. Strikingly, the rAg5'NTE hydrolyzed ADP like a *Cimex*-type apyrase in the presence of $Ca^{2+}$ only (Fig. 2d) despite being phylogenetically distant (Fig. S2a). The amino acid residues binding to calcium are well conserved amongst the members belonging to both 5' nucleotidase type and *Cimex* type apyrases (Fig. S2b). Salivary apyrases from hematophagous insects inhibit ADP-mediated platelet activation and aggregation which could have significant implications in human hemostatic responses during insect feeding[32]. We observed that rAg5'NTE (AGAP011026-PA) strongly inhibits ADP-mediated platelet aggregation in vitro (Fig. 2e). These results confirm that AGAP011026-PA (Ag5'NTE) is the *An. gambiae* salivary apyrase, henceforth referred to as AgApyrase (AgApy).

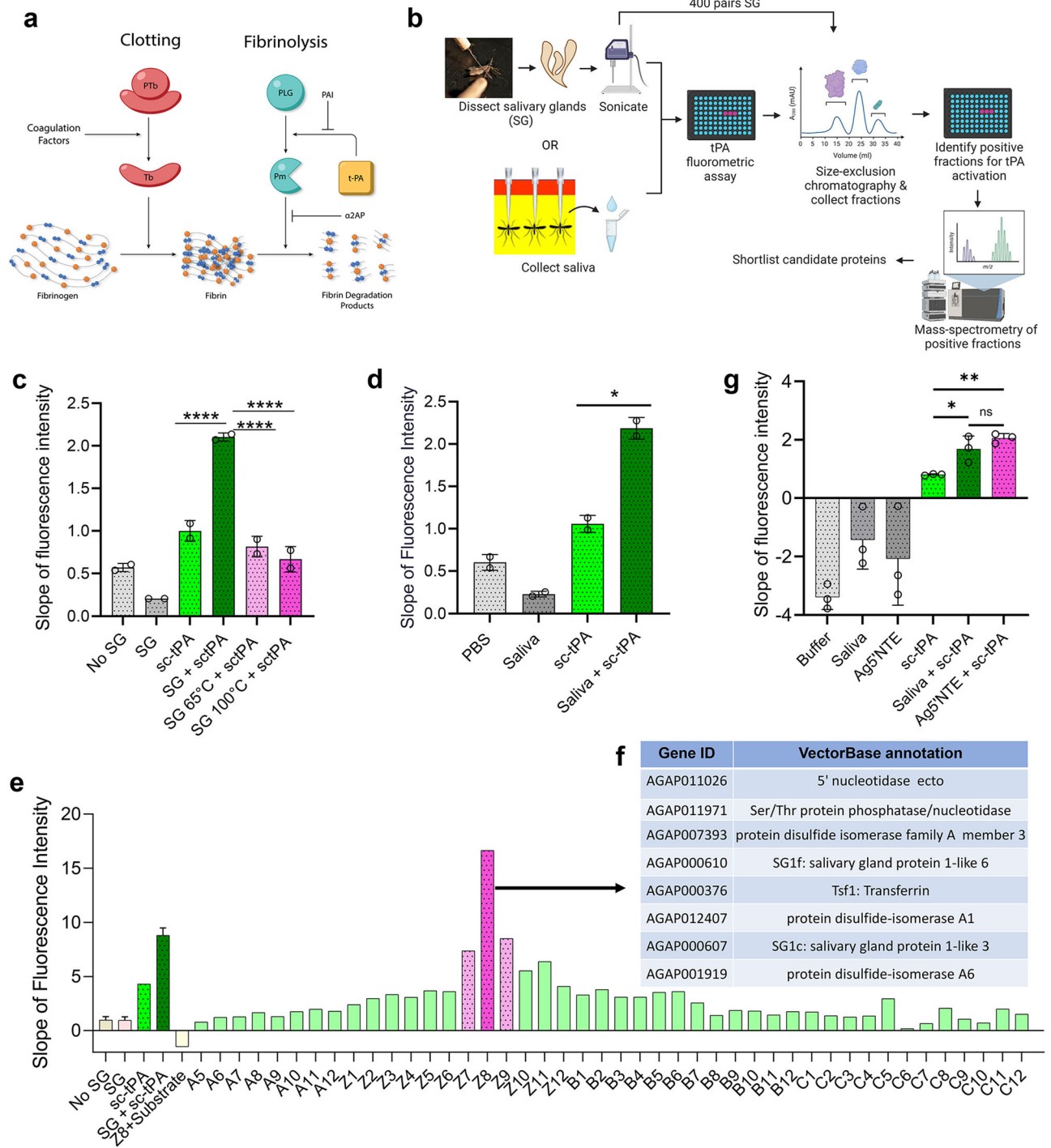

**Fig. 1 | *An. gambiae* salivary protein AGAP011026 activates tPA. a** Diagram summarizing clotting and fibrinolysis. PTb: pro-thrombin, Tb: thrombin, PLG: plasminogen, Pm: plasmin, PAI: plasminogen activator inhibitor, t-PA: tissue plasminogen activator, α2AP: α−2 antiplasmin. **b** Workflow diagram to identify the mosquito salivary protein activating tPA. Figure 1b was created with BioRender.com released under a Creative Commons Attribution-NonCommercial-NoDerivs license. **c**, **d** Fluorogenic assay for single-chain tPA (sc-tPA) activation reveals that mosquito salivary gland extracts (SG) (**c**) and saliva (**d**) activate sc-tPA. Heating the extract at 65 °C or 100 °C prevents tPA activation. $n = 2$, each performed in duplicates. One-way ANOVA with Šídák's multiple comparisons test, $*P = 0.0214$, $****P < 0.0001$. **e** Fractions from *An. gambiae* salivary gland extracts obtained by size-exclusion chromatography were tested for tPA activation. Representative experiment. $n = 1$. **f** Mosquito salivary proteins shortlisted as potential tPA activators. **g** *An. gambiae* salivary 5′ nucleotidase ecto (Ag5′NTE) identified as the saliva tPA activator. $n = 3$. One-way ANOVA with Šídák's multiple comparisons test, $*P = 0.0226$, $**P = 0.0042$. Error bars ± S.E.M. Source Data are provided as a Source Data file.

## AgApyrase interacts with human tPA

To determine whether AgApyrase and human tPA interact, we performed ELISA overlay assays, which showed that recombinant AgApyrase directly interacts with tPA (Fig. 2f, pink bar). The negative controls included buffer or AgApyrase without tPA, and the unrelated protein AGAP007393 with and without tPA (Fig. 2f). AGAP007393, also known as ZPJVR3, is one of the eight tPA activator candidates and after Ag5′NTE, AGAP007393 showed a slightly higher slope of fluorescence intensity for tPA activation when compared to all the remaining proteins in consideration (Fig. S1b). We further studied the binding of

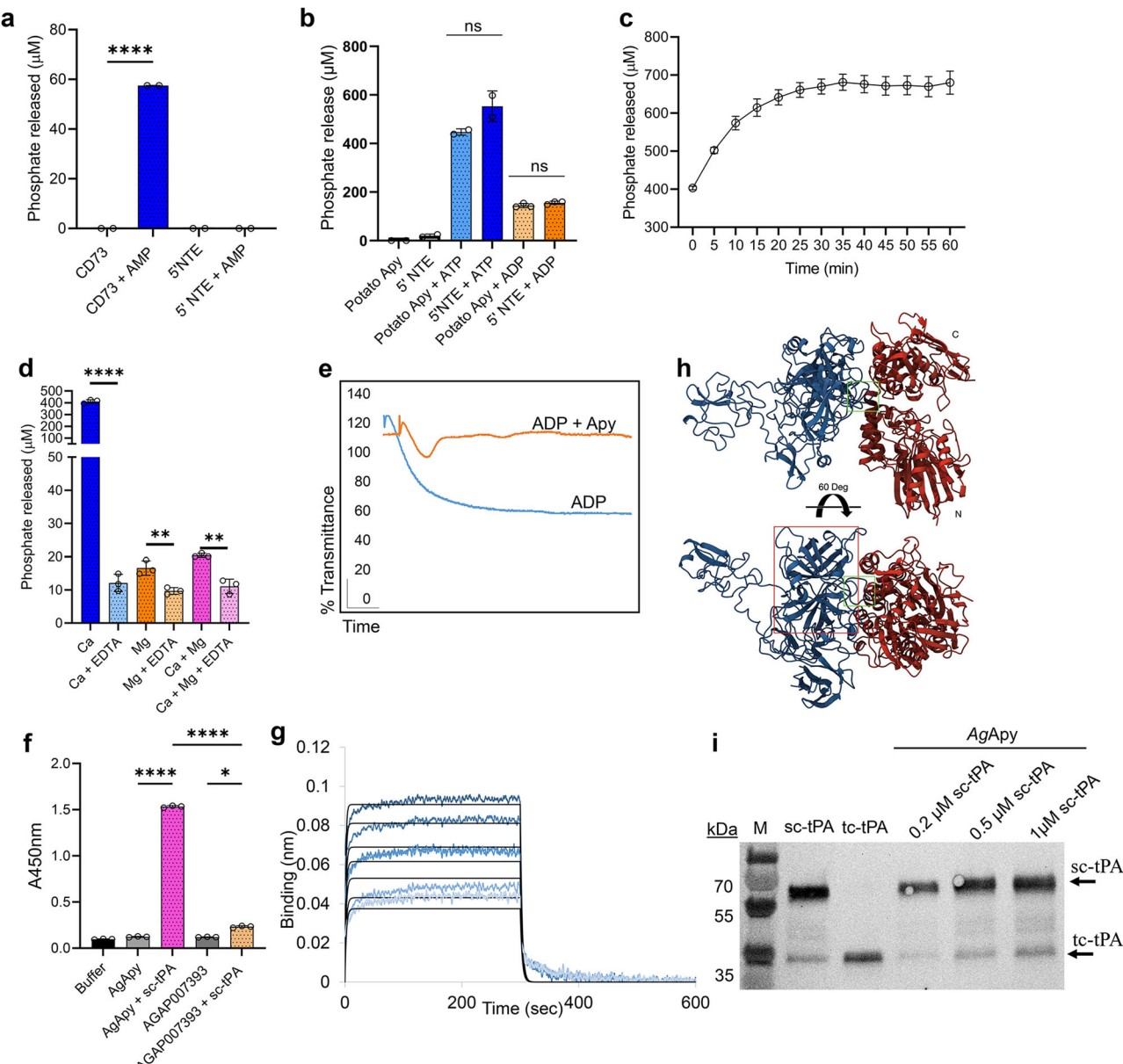

**Fig. 2 | The salivary tPA activator AGAP011026 is a mosquito salivary apyrase.**
**a** 5' nucleotidase ecto activity was measured by the release of inorganic phosphate from AMP using the malachite green assay. $n = 2$, each performed in duplicates. Two-tailed unpaired $t$-test, ****$P < 0.0001$. **b** AGAP011026 is an apyrase. Inorganic phosphate release from ATP or ADP was measured using Fiske Subbarow reagent. $n = 2$, each performed in duplicate ns: not significant. Two-tailed unpaired $t$-test, ns: not significant. **c** Time course of ADP hydrolysis by AgApyrase. Data shown from one representative experiment ($n = 3$) performed in duplicates. **d** Apyrase activity of AGAP011026 was measured using malachite green assay in presence of either 5 mM CaCl2 and/or 5 mM MgCl2 with or without addition of 5 mM EDTA. Representative experiment performed in triplicates. Two-tailed unpaired $t$-test, ****$P < 0.0001$, **$P = 0.0075$ for Mg vs Mg+EDTA, and 0.0020 for Ca vs Ca + Mg + EDTA. **e** rAgApyrase inhibits the platelet aggregation induced by ADP (orange) in contrast to the control buffer with ADP (blue) which aggregated all the available platelets, measured by light transmittance. Representative experiment.
**f** Interaction of tPA with Ag5'NTE using ELISA overlays. Wells were coated with rAgApyrase and then incubated with or without sc-tPA. Anti-tPA antibodies were used to detect sc-tPA binding. A well was coated with an unrelated protein

AGAP007393 as negative control. One-way ANOVA with Dunnet's multiple comparison test, ****$P < 0.0001$; *$P < 0.0109$. **g** Biolayer interferometry experiments demonstrate the in vitro interaction between biotinylated AgApyrase immobilized on SA biosensors and 250-1000 nM sc-tPA. Data and the 1:1 fit curves are shown in blue and black, respectively. Representative experiment. **h** Model of the AlphaFold2 and OmegaFold predicted interaction of AgApyrase with human tPA in two orientations, with Apyrase in red (with N and C lobes labeled) and tPA in blue. The buried surface area for the Apyrase-tPA complex final model is 1467 A2. The interaction between the two molecules is mediated by one of tPA the protruding loops (1006-HEALSP-1011, indicated with a green box) in the tPA serine protease domain, which is recognizable and marked by a red box. **i** Western blot analysis of sc-tPA after incubation with rAgApyrase. The top arrow indicates sc-tPA (~ 65 kDa) and the bottom arrow indicates tc-tPA (~ 32 kDa) positive controls. No extra cleaved bands or an increase in sc-tPA cleavage are seen in any of the sc-tPA samples incubated with rAgApyrase. M: molecular marker. Error bars indicate SD in (**a**–**d**) and (**f**) indicate standard deviation; and SEM in panel **c**. Source Data are provided as a Source Data file.

apyrase to tPA using biolayer interferometry (BLI) (Fig. 2g). The BLI data confirmed the binding of apyrase and tPA as seen in the ELISA. In silico structural analysis of the interaction between AgApyrase and human tPA generated using AlphaFold2 and OmegaFold produced a complex model with a buried surface area 1467 A$^2$, where the interaction between the two molecules is mediated by one of the protruding loops in the tPA serine protease domain (Fig. 2h). The AgApyrase residues mediating the interaction are mostly located in the nucleotide binding site and the nucleosidase catalytic site. It is worth noticing that the residues involved in the Apyrase-tPA contact are highly conserved across the selected set (Fig. S2c). In mammals, activation of single-chain tPA (sc-tPA) takes place by proteolytic cleavage by plasmin generating two-chain tPA (tc-tPA), or by allosteric interaction with other proteins like fibrin[33,34]. The sc-tPA is a ~ 65 kDa protein and it results in smaller ~32 kDa tc-tPA fragments upon activation. Incubation of rAgApyrase with sc-tPA did not enhance its cleavage into two-chain tPA (tc-tPA) (Fig. 2i), suggesting an allosteric activation mechanism. The faint tc-tPA band obtained at ~32 kDa in sc-tPA (Fig. 2i, lane 2, sc-tPA) is the result of the trace amount of tc-tPA present in the commercial sc-tPA which is 85% pure (Innovative Research #IHUTPA85SC1MG).

## Mosquito saliva is mixed with the ingested blood

Our study builds on our previous finding that fibrin polymerizes in the mosquito midgut and increases the viscosity of the blood meal, serving as a barrier that hampers parasite motility and infectivity[3]. Therefore, clotting factors are active in the midgut blood bolus. Since we showed that salivary AgApyrase activates the fibrinolytic system in vitro (Fig. 1), we postulated that ingestion of AgApyrase enhances fibrin degradation, thereby reducing the blood meal viscosity and boosting parasite infection in the mosquito midgut. To confirm the presence of salivary apyrase in the mosquito blood bolus, we performed immunohistochemistry with anti-AgApyrase antibodies (Fig. S3a) on blood (human)-fed mosquito midguts dissected 30 min post feeding and on unfed whole mosquitoes. We observed that *An. gambiae* mosquitoes ingest a significant amount of apyrase during feeding (Fig. 3a, Fig S3b), which is supported by other studies showing the ingestion of saliva during blood feeding[4,13,35]. However, our study demonstrates the extent of saliva ingestion and its distribution within the blood meal. AgApyrase could be detected in smaller quantities in the lumen of unfed mosquitoes (Fig. S3c), presumably from saliva ingested without feeding on blood since this apyrase is not expressed in mosquito midguts, either before or after blood feeding, based on qPCR analysis (Fig. S3d) and according to published transcriptomes included in VectorBase[36–38]. Further, western blot analysis showed that the anti-AgApyrase antibodies only reacted with the homogenate of one blood fed mosquito midgut and did not react with human plasma or serum (Fig. S3e). This clearly demonstrates that the source of the apyrase observed in the midgut blood bolus (Fig. 3a) is mosquito saliva and not the human blood utilized for feeding the mosquitoes. The ingestion of salivary proteins by the mosquito while taking a blood meal was also confirmed by performing a similar experiment with an antibody against anopheline anti-platelet protein (AAPP), another mosquito salivary protein. We observe AAPP ingestion during a blood meal (Fig. S3f). Similar to apyrase, AAPP is not expressed in the mosquito midgut epithelium either prior or after blood ingestion[36–38]. The difference in intensity between apyrase and AAPP can be explained by the fact that apyrase antibodies are polyclonal against the whole recombinant protein, whereas the antibody against AAPP was developed against a 20 amino acid peptide. Therefore, anti-AgApyrase antibodies recognize more binding sites on AgApyrase than anti-AAPP antibodies on AgAAPP, giving a stronger signal. These results confirm that, when taking a blood meal, a significant amount of saliva, especially salivary AgApyrase, is ingested by the mosquito. This finding has important implications for the role of saliva in modulating hemostatic and inflammatory responses in the mosquito blood bolus, areas that deserve further investigation due to the potential effect on vectorial capacity and pathogen transmission.

## Apyrase enhances fibrinolysis in the mosquito midgut

Next, we tested whether AgApyrase enhanced fibrin degradation in the midgut blood bolus. To determine whether apyrase ingestion increased fibrin degradation, we measured the concentration of D-dimers, degradation products of cross-linked fibrin, formed in the midgut blood bolus of mosquitoes fed on mice before and after intravenous rAgApyrase injection. Results from a competitive ELISA assay showed a significant increase in D-dimers in mosquitos fed on rAgApyrase supplemented blood, showing that apyrase ingestion enhanced fibrinolysis in the blood bolus (Fig. 3b). Further quantification was performed using immunohistochemistry on sections obtained from mosquitoes fed on mice before and after AgApyrase injection with anti-fibrinogen antibody (Fig. 3c and S4). At 30 min post feeding, we observed a significant decrease of 24% in the mean signal, and a decrease of 25% in the percentage area of fibrinogen/fibrin staining in midguts with AgApyrase (Fig. 3d), supporting a faster degradation of fibrin in the presence of AgApyrase. Using scanning electron microscopy (SEM), we observed the fibrin networks in the midgut blood bolus of mosquitoes that fed on mice before and after intravenous injection of rAgApyrase. Dissection at 30 min or 4 h post-feeding showed that midguts from control mosquitoes (before apyrase injection) exhibited an organized fibrin polymer network, whereas those fed on rAgApyrase injected mice displayed a sponge-like mesh (Fig. 3e, Figs. S5 and S6). Immune transmission electron microscopy of mosquito blood boluses confirmed the presence of fibrinogen in the space between red blood cells (Fig. S7), supporting that the observed fibers in transmission electron microscopy are fibrin. Further, SEM of blood boluses from mosquitoes fed on rAgApyrase immunized mice show complex thick fibers organized in a tight network when compared to mosquitoes fed on the adjuvant control group (Fig. 3f, Fig. S8). In conclusion, our results demonstrate that salivary apyrase ingested by *An. gambiae* mosquitoes during blood feeding increases fibrinolysis in the midgut.

## Apyrase inhibits platelet activation and aggregation in the mosquito midgut

Platelets and fibrin play crucial roles in coagulation. Platelets initiate the formation of a primary platelet plug at injury sites, whereas fibrin stabilizes it[39]. The release of ADP from activated platelets during coagulation further amplifies platelet aggregation[40]. Some hematophagous arthropods such as mosquitoes can inhibit ADP-mediated platelet aggregation through their salivary apyrase[41]. Our study demonstrated that recombinant apyrase from *An. gambiae* mosquitoes (rAgApyrase) inhibits ADP-mediated platelet aggregation in vitro (Fig. 2e). We further showed that rAgApyrase also inhibited human whole blood coagulation in vitro (Fig. S9). To investigate the effect of salivary apyrase on platelet activation and aggregation in the mosquito midgut, we performed immunohistochemistry on midgut sections obtained from mosquitoes fed on mice with or without intravenous rAgApyrase supplementation and used P-selectin (CD62P) staining as a marker for platelet activation. Our results show a significant reduction in P-selectin staining (dark-brown spots) in the midguts of mosquitoes fed on rAgApyrase supplemented mice when compared to the control mice (Fig. 4a, Fig. S10). Quantification of P-selectin staining also showed a significant decrease of 30% in both the mean signal and percentage area of staining for midguts obtained from mosquitoes fed on rAgApyrase supplemented mice (Fig. 4b). Scanning electron microscopy images of mosquitoes fed on rAgApyrase supplemented mice showed platelet aggregates that were easily distinguishable (Fig. 4c), whereas in the rAgApyrase treated group, platelet aggregates were rarely found and when observed, they were mostly present at the

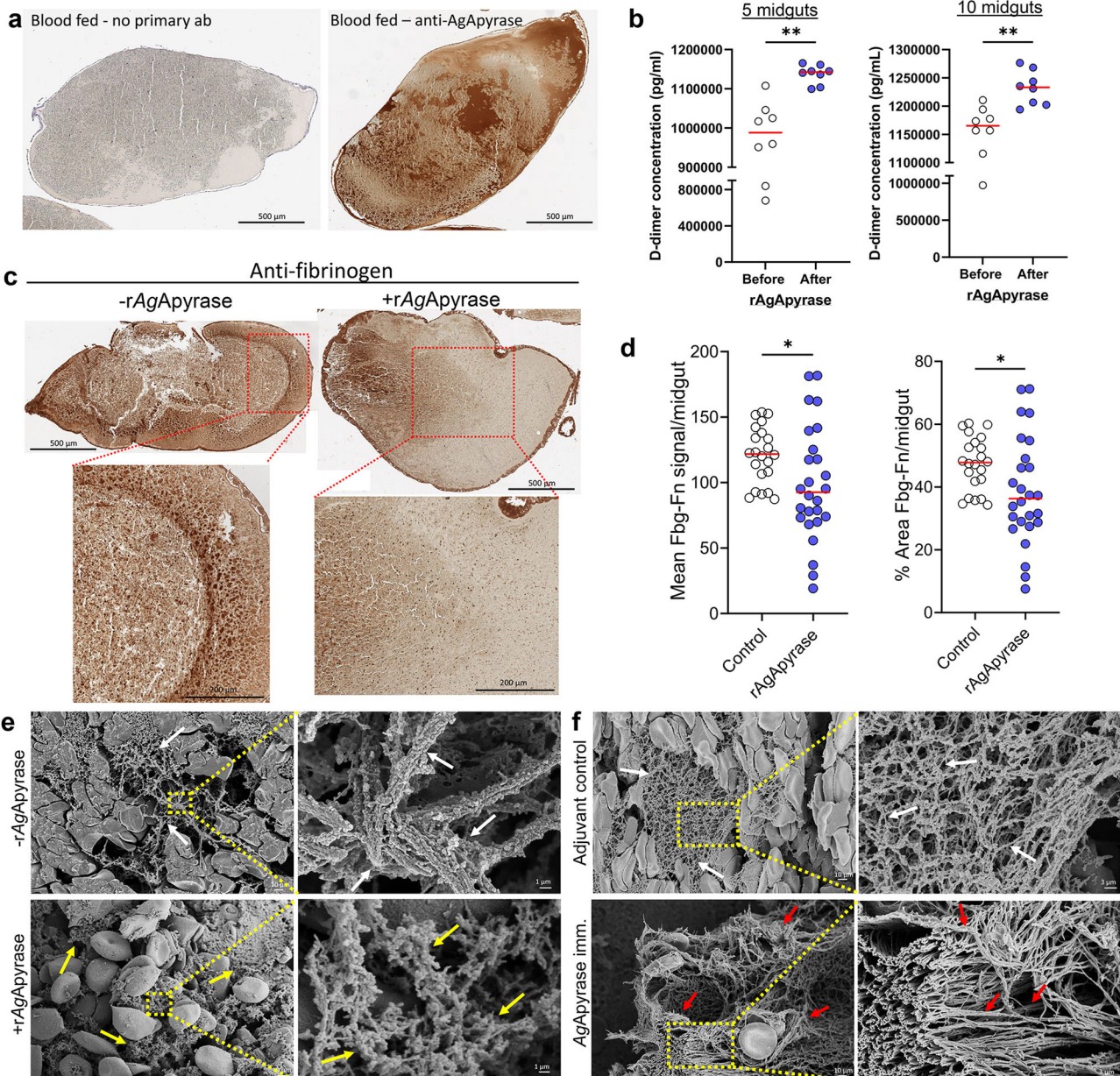

**Fig. 3 | Salivary AgApyrase is ingested during blood feeding and enhances fibrin degradation in the blood bolus. a** Immunohistochemistry performed on blood fed mosquito midgut shows ingestion of salivary AgApyrase in the mosquito midgut (dark brown patches on the right panel) by staining with anti-apyrase antibodies. Blood fed mosquito midgut with no primary antibody did not show any signal. **b** Supplementation of the blood meal with rAgApyrase increases D-dimer formation. Mosquitoes were fed on mice before or after intravenous injection of rAgApyrase, and midguts were dissected 30 min post feeding to measure D-dimer formation. Each dot represents a pool of 5 (left) or 10 (right) midguts. $N = 4$, two replicates each. Two-tailed unpaired $t$-test, **$P = 0.0021$ for 5 midguts and 0.0012 for 10 midguts. **c** Immunohistochemistry was performed on mosquito blood boluses before and after supplementation with rAgApyrase and stained with an anti-fibrinogen antibody (labels fibrinogen and fibrin). **d** Quantification of the percentage area (left) and the mean signal (right) of fibrinogen (Fbg)/fibrin (Fn) per midgut (additional representative IHC images shown in Fig. S4). Each dot represents an individual midgut. Data from two independent experiments, $n = 22$ and 26 midguts for control and rAgApyrase, respectively. Two-tailed Mann Whitney test, ***$P = 0.0001$. **e** Scanning electron microscopy (SEM) of blood boluses from *An. gambiae* female mosquitoes fed on mice before or after intravenous injection of rAgApyrase. Note the well organize fibers formed in the blood bolus before the supplementation with rAgApyrase (white arrows) as compared to the sponge-like structure formed after rAgApyrase supplementation (yellow arrows). Yellow dotted pattern shows the region magnified in adjacent panels. **f** SEM of blood boluses from mosquitoes fed on rAgApyrase immunized mice. Control mosquitoes were fed on mice treated with adjuvant. Midguts were dissected 30 min post feeding. Note the fiber organization and thickness (red arrows) observed in boluses from mosquitoes fed on rAgApyrase immunized mice as compared to the normal fibers (white arrows) from mosquitoes fed on adjuvant treated mice. Representative images (**e**, **f**) from a single experiment at two time points (30 min and 4 h post infection). Source Data are provided as a Source Data file.

4 h timepoint (Fig. 4c, Fig. S6Gi and S6Gii). Interestingly, when mosquitoes were fed on rAgApyrase immunized mice, we observed platelet aggregates that included platelet fragmentation into smaller vesicles (Fig. 4d, and Fig. S8l–n). These structures resemble a phenomenon previously described in an in vivo model for thrombus formation where the tightest regions of the thrombus were composed of highly activated and tightly packed platelets that fragmented into vesicular bodies[42].

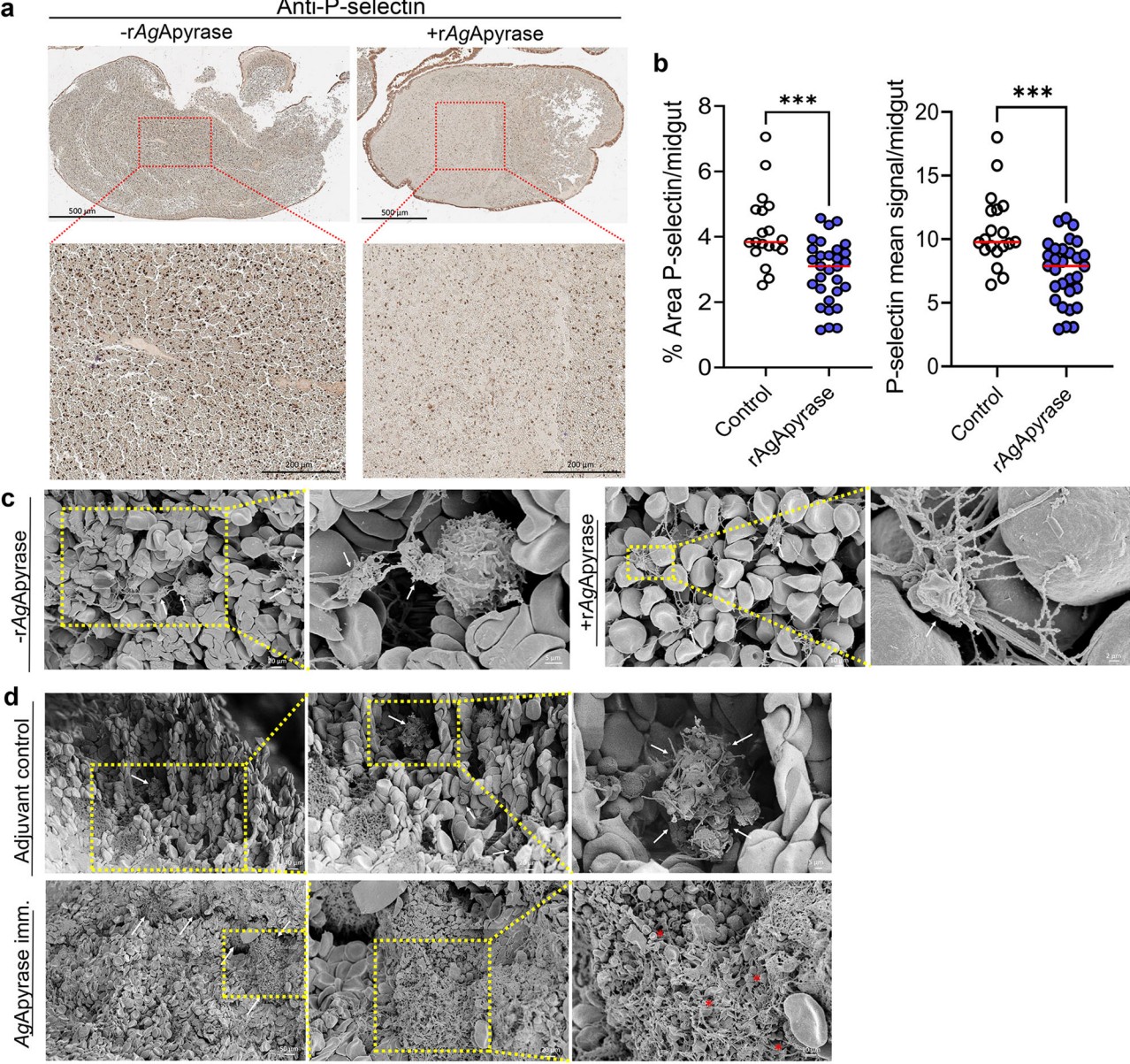

**Fig. 4 | Salivary AgApyrase inhibits platelet activation and aggregation in the blood bolus. a** Immunohistochemistry was performed on mosquito blood boluses before and after supplementation with rAgApyrase and stained with P-selectin (a marker for platelet activation). **b** Quantification of the percentage area and the mean signal of P-selectin staining per midgut (additional representative IHC images shown in Fig. S9). Each dot represents an individual midgut. Data pooled from two independent experiments, $n = 19$ and 30 midguts for control and rAgApyrase, respectively. Mann Whitney test, ***$P = 0.0001$. **c, d** SEM showing platelet aggregation in blood boluses before and after supplementation with rAgApyrase (**c**) or from mosquitoes fed on adjuvant or rAgApyrase immunized mice (**d**). White arrows indicate the aggregation of platelets. Red asterisks show fragmentation of platelets into vesicular bodies reminiscent of hypercoagulation. Representative images from duplicate experiments at two time points (30 min and 4 h post infection). Source Data are provided as a Source Data file.

## Apyrase regulates neutrophil activation in the mosquito midgut

In humans, immunothrombosis involves the cooperative action of the immune and coagulation systems to target invading pathogens. Neutrophil extracellular traps (NETs) play a role in this process by trapping and killing pathogens through the release of decondensed chromatin, histones, and various neutrophil granule proteins, including myeloperoxidase and elastase. Activated platelets and fibrin induce NET formation and, in a positive feedback loop, the negatively charged NETs can activate the contact phase of blood coagulation, promoting further activation of platelets and enhancing fibrin polymerization[43,44]. NETs have been implicated in thrombosis in homeostasis and disease. We investigated the influence of AgApyrase on NETosis in the presence of platelets. We found that rAgApyrase, in the presence of platelets, significantly reduces NET formation by 79% (Fig. 5a, b). We also observed that rAgApyrase reduces the formation of total cell- and mitochondrial-derived reactive oxygen species (ROS) in neutrophils, which are considered fundamental for most forms of NET formation (Fig. 5c, d)[45]. Additionally, immunohistochemistry analysis showed a significant decrease in the percentage area (50%) and mean signal (45%) for extracellular neutrophil elastase staining, a molecule extruded in the NETs and hence a marker for NETs, in the mosquito blood bolus after rAgApyrase supplementation (Fig. 5e, f and Fig. S11). These findings suggest that salivary apyrase inhibits neutrophil activation in the mosquito midgut.

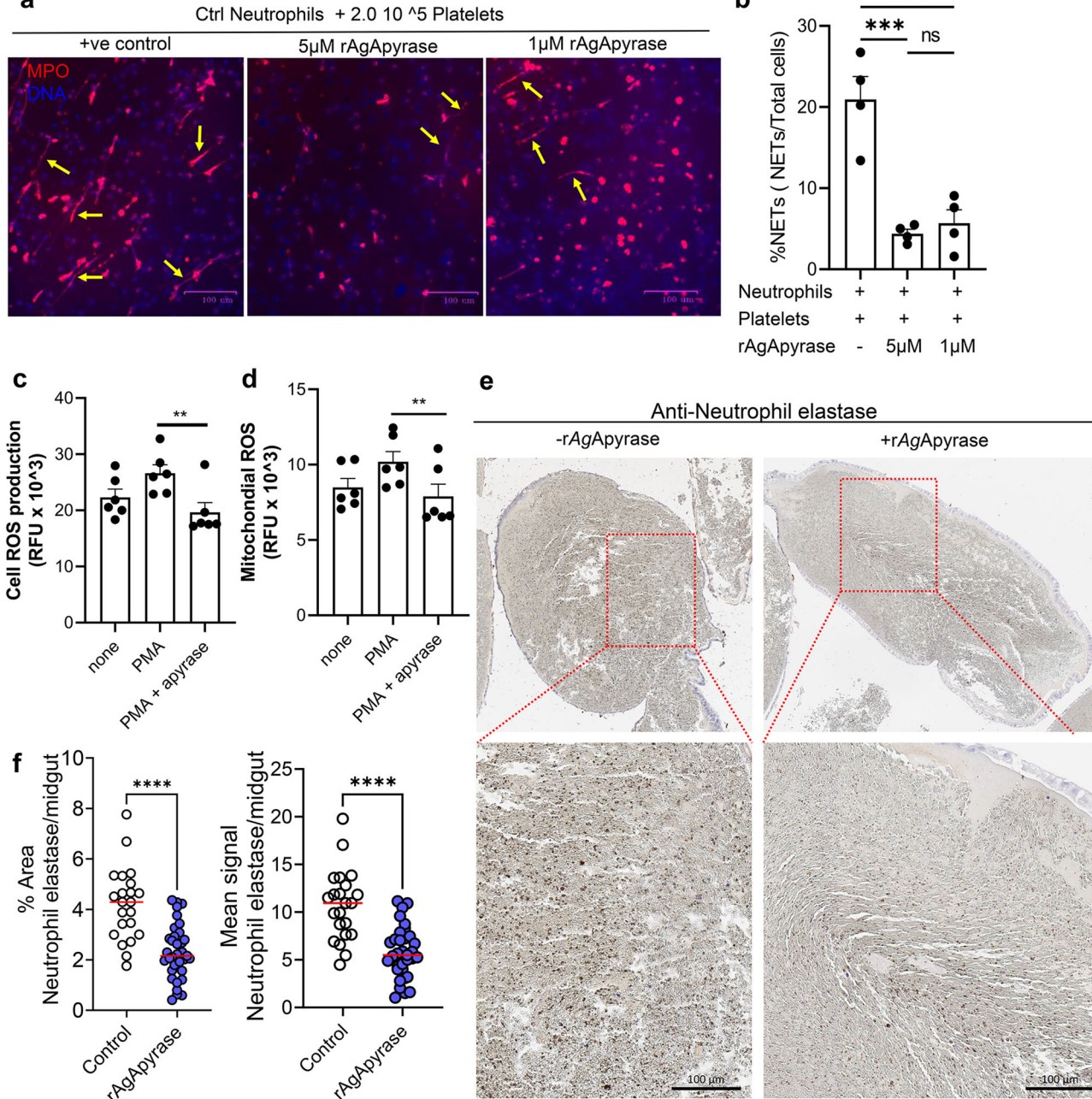

**Fig. 5 | AgApyrase inhibits platelet-mediated NET formation. a, b** Neutrophils were incubated with platelets in the presence or absence of rAgApyrase. NETs were quantified by immunofluorescence microscopy using an anti-MPO (myeloperoxidase) antibody (**a**). DNA was stained with DAPI. Yellow arrows point released neutrophil DNA stained with MPO. The percentage of NETs (**b**) was calculated as an average of 5–10 fields (400X) normalized to total number of neutrophils. Results expressed as mean % ± SEM. $n = 4$. One-way ANOVA with Šídák's multiple comparisons test, ***$P = 0.0006$ for control vs 5 µM and 0.001 for 10 µM. **c, d** AgApyrase reduces neutrophil ROS production. Whole cell (**c**) or mitochondrial (**d**) ROS production in neutrophils activated or not with phorbol myristate acetate (PMA) and incubated in the presence or absence of rAgApyrase. One-way ANOVA with Friedman test. **$P = 0.003$. $n = 6$. Error bars indicate SEM. **e** Immunohistochemistry was performed to quantify neutrophil elastase as a marker for NETosis. **f** Quantification of the percentage area and the mean signal of neutrophil elastase staining per midgut (additional representative IHC images shown in Fig. S10). Data pooled from two independent experiments. Each dot represents an individual midgut. $n = 22$ and 35 midguts for control and rAgApyrase, respectively. Mann Whitney test, ****$P = 0.0001$. Source Data are provided as a Source Data file.

## Apyrase drives parasite transmission to mosquitoes

In this study, we showed that salivary apyrase from *An. gambiae* enhances fibrinolysis and inhibits platelet activation and aggregation in the midgut. We previously demonstrated that fibrin polymerization increases the viscosity of the blood bolus, which in turn reduces parasite motility and transmission to the mosquito[3]. Therefore, we hypothesize that salivary apyrase enhances parasite transmission to the mosquito by reducing the blood bolus viscosity.

To test this hypothesis, we performed passive administration of rAgApyrase on mice infected with *Plasmodium berghei* parasites. A group of *An. gambiae* female mosquitoes were fed on the infected mouse before (control group) and after rAgApyrase intravenous injection. Supplementation of the infected mouse blood with rAgApyrase significantly increased the number of oocysts in the mosquito, while heat-denatured recombinant protein failed to increase the infection (Fig. 6a and S12). Furthermore, immunization

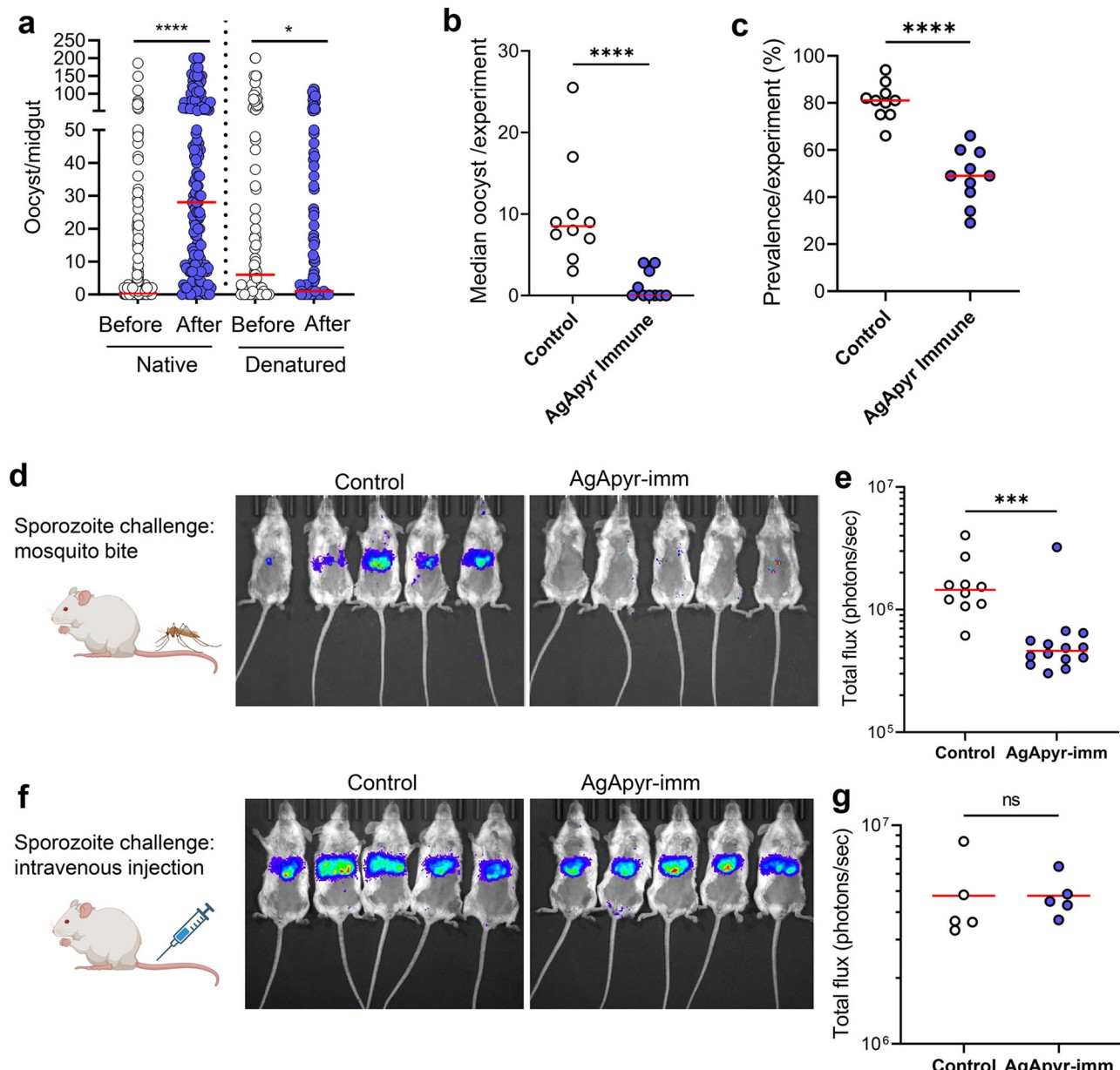

**Fig. 6 | Effect of *AgApyrase* on *P. berghei* transmission. a** AgApyrase facilitates *P. berghei* infection of mosquito midguts. Oocyst numbers from midguts of *An. gambiae* mosquitoes that fed on a *P. berghei* infected mouse before or after the intravenous injection of native or heat-denatured rAgApyrase. Data pooled from three individual experiments shown in Fig. S10, and groups were compared with two-tailed *t*-test followed by Two-tailed Mann-Whitney comparison test. *n* = 168, 158, 105 and 117 midguts for before and after in native and denatured, respectively. Red lines indicate median. ****P < 0.0001; *P = 0.0456. **b, c** AgApyrase immunization inhibits *P. berghei* midgut infection. Oocyst numbers (**b**) and infection prevalence (**c**) were determined in the midguts of *An. gambiae* mosquitoes fed on *P. berghei* infected BALB/c mice previously immunized with rAgApyrase using Magic Mouse adjuvant or with adjuvant alone as control. Each dot represents median oocyst number (**b**) or prevalence (**c**) from mosquitoes fed on one mouse. *n* = 10 mice per group. Groups were compared with two-tailed *t*-test followed by two-tailed Mann-Whitney comparison test. Red lines indicate median. ****P < 0.0001. The oocysts numbers from mosquitoes feeding on each individual mouse are shown on Fig. S11. **d, e** AgApyrase immunization inhibits sporozoite transmission. BALB/c mice immunized with rAgApyrase in Magic Mouse adjuvant or adjuvant alone (control) were challenged with the bite of five *An. stephensi* mosquitoes infected with *P. berghei* sporozoites expressing the luciferase gene. Sporozoite infectivity was determined by measuring luciferase activity in the mouse liver 42 h post challenge. Luminescence signal in the mice livers is shown in panel (**d**) and the quantification in panel (**e**). Data pooled from two independent experiments and groups were compared with two-tailed *t*-test followed by two-tailed Mann–Whitney comparison test. *n* = 10 and 14 mice for control and AgApyr-imm, respectively. Red lines indicate median. ***P = 0.0002. **f, g** Similar experiment as in (**d, e**), but mice were challenged with 5000 sporozoites injected intravenously. Data from a single experiment, *n* = 5 mice per group. Two-tailed Mann-Whitney comparison test. Source Data are provided as a Source Data file. Figure 6d and f were created with BioRender.com released under a Creative Commons Attribution-NonCommercial-NoDerivs license.

of mice with rAgApyrase reduced the oocyst intensity and infection prevalence in mosquitoes when compared to mice immunized with Magic™ Mouse adjuvant alone (Figs. 6b, c, and S13). The median oocyst numbers in mosquitoes fed on rAgApyrase-immunized mice ranged between 0 to 4, compared to 3 – 25 in control mice

(Fig. 6b and S13). The median infection prevalence was reduced from 81% in control mice to 49% in rAgApyrase-immunized mice (Fig. 6c). Our results show that salivary apyrase ingested by the mosquito during blood feeding increases parasite infectivity to the mosquito.

### Salivary apyrase facilitates sporozoite transmission

We further studied whether salivary *Ag*Apyrase is required for sporozoite transmission from the mosquito to the mammalian host. Mice immunized with r*Ag*Apyrase were challenged with the bite of five mosquitoes infected with transgenic *P. berghei* sporozoites expressing the reporter luciferase gene. Analysis of the parasite's luciferase signal in the mouse liver 40 h post-challenge served as an indicator of sporozoite infectivity (escape from the skin and liver invasion) and exoerythrocytic form development[46]. We observed a significant reduction of 68% in luminescence in mice immunized with rAgApyrase compared to adjuvant treated mice (Fig. 6d, e). This reduction resulted in a delayed time to patency of 1.5 days (Fig. S13e). Notably, when sporozoites were injected intravenously to bypass the skin barrier, the parasite liver load between adjuvant and rAgApyrase immunized mice remained unchanged (Fig. 6f, g), suggesting that rAgApyrase immunization affects sporozoite infectivity in the skin. These findings show that salivary AgApyrase is an important factor for successful sporozoite transmission via the mosquito bite.

## Discussion

Our research sheds light on the crucial role of mosquito salivary apyrase in regulating hemostasis and promoting *Plasmodium* development in the mosquito midgut and sporozoite transmission during a mosquito bite. These findings emphasize the role of saliva proteins during malaria parasite transmission and identify salivary apyrase as a unique candidate for prophylactic and transmission-blocking vaccine, targeting transmission from the vector to the human and from the human to the mosquito vector. Vector saliva regulates hemostatic responses and pathogen transmission at the bite site, however the understanding of how the vector saliva regulates these responses in the mosquito blood bolus and its impact on *Plasmodium* transmission is limited[35].

We previously showed that activation of the coagulation system in the mosquito midgut increases the viscosity of the blood bolus, which serves as a barrier that reduces parasite infectivity to the mosquito[3]. The parasite overcomes this barrier by hijacking plasminogen activators (tPA and uPA) and plasminogen that is activated into plasmin, a protease that degrades fibrin and facilitates parasite motility in the blood bolus[3]. Here, we show that mosquito salivary apyrase is ingested in high quantities during blood feeding, activates fibrinolysis through the activation of tPA, reduces coagulation by inhibiting platelet activation and aggregation, and inhibits NET formation in the blood bolus. Therefore, salivary apyrase is an important player in preventing hemostasis and inflammation in the blood bolus, properties that will facilitate blood digestion, enhance mosquito fitness and therefore, enable the establishment of parasite infection in the mosquito midgut (Fig. 7). Our studies were performed with *P. berghei* parasites as it allows for natural transmission from the host to the mosquito vector. The standard membrane feeding assay (SMFA) is the conventional method used to evaluate transmission-blocking (TB) activity for *P. falciparum*. However, a limitation of this system is its failure to replicate the conditions present during natural parasite transmission, including the involvement of the coagulation system and immune cells. Despite this limitation, we showed that the recombinant mosquito salivary apyrase inhibited human blood coagulation in vitro, suggesting a similar mode of action for *P. falciparum* parasite in vivo. Although our previous work using transgenic mosquitoes that inhibit fibrinolysis previously showed the relevance of this pathway for transmission of *P. falciparum* and *P. vivax*, future studies to evaluate the effect of apyrase and anti-apyrase antibodies in the transmission of *P. falciparum* and other human malaria parasites will require the establishment of an in vivo model that includes hemostatic and inflammatory responses.

In our previous publication, we demonstrated that *Plasmodium* sporozoites utilize tissue plasminogen activator (tPA) to activate plasminogen bound to their surface[3]. The resulting surface-bound plasmin facilitates the degradation of extracellular matrix proteins, thereby promoting sporozoite motility through the matrix both in vitro in Matrigel and in vivo in the dermis of a mouse. Building upon these findings, we propose that salivary apyrase, co-injected with sporozoites into the dermis by the mosquito, activates sporozoite-bound tPA, thus accelerating the conversion of sporozoite-bound plasminogen into active plasmin. Therefore, it is conceivable that apyrase enhances sporozoite motility and facilitates their escape from the skin by promoting extracellular matrix degradation in the dermis through activation of the fibrinolytic system. However, further experiments are required to validate this hypothesis. Nonetheless, our findings underscore the critical role of mosquito salivary apyrase in facilitating sporozoite transmission to the mammalian host.

Blood feeding by mosquitoes is a fast process compared to other hematophagous vectors, taking only a few minutes. Although *Anopheles* mosquitoes can feed from blood pools, they feed more frequently by cannulating venules or arterioles[47]. The benefits of saliva at the bite site are more apparent during pool feeding compared to direct vessel feeding. Mosquito saliva is primarily assumed to exert its anti-hemostatic and anti-inflammatory effects on the host skin during probing. Interestingly, studies have shown that saliva is expelled throughout the entire blood feeding process in Anopheline mosquitoes, ensuring a constant flow of saliva and salivary proteins[48]. A previous study using *An. campestris* mosquitoes found that the most abundant salivary proteins, including apyrase, were substantially depleted after ingestion of a blood meal, showing that a significant amount of saliva is delivered during blood feeding[49]. Our results show that during blood feeding, a substantial amount of the expelled saliva is ingested and plays a major role in regulating hemostasis in the blood bolus and facilitating *Plasmodium* development in the mosquito.

Our findings underscore the importance of investigating the diverse roles assigned to mosquito saliva, not only during probing and feeding on the host skin but also within the mosquito blood bolus, where it can significantly impact parasite transmission. Studying the functional roles of saliva proteins within this microenvironment can reveal critical insights into the complex interactions between the mosquito, host, and parasite during transmission paving the way for the development of strategies for blocking the transmission of not only malaria but also other vector-borne diseases.

## Methods

### Ethical statement

This research complies with all relevant ethical regulations. The study was performed in strict accordance with the recommendations from the Guide for Care and Use of Laboratory Animals of the National Institutes of Health (NIH). The animal use was done in accordance with the Use Committee or The NIH Animal Ethics Proposal SOP LMVR 22.

Platelet rich plasma and whole blood in acid citrate dextrose was obtained from normal healthy donors enrolled in a protocol approved by the National Institutes of Health Clinical Center Institutional Review Board (NIH protocol 99-CC-0168 "Collection and Distribution of Blood Components from Healthy Donors for In Vitro Use"). Blood donors provided written informed consent, and platelets were de-identified prior to distribution.

### Mosquito rearing and *Plasmodium berghei* infection

*Anopheles gambiae* G3 and *An. stephensi* (Nijmegen) mosquitoes were reared at 27 °C and 80% humidity with a 12-h light/dark cycle under standard laboratory conditions and maintained with 10% Karo syrup solution during adult stages. *P. berghei* infections were performed using a transgenic line expressing mCherry and luciferase[50] and maintained by serial passages in female Swiss webster mice. Parasitemia was assessed by light microscopy by using methanol-fixed blood-smears from infected mice and stained with 10% Giemsa. Female

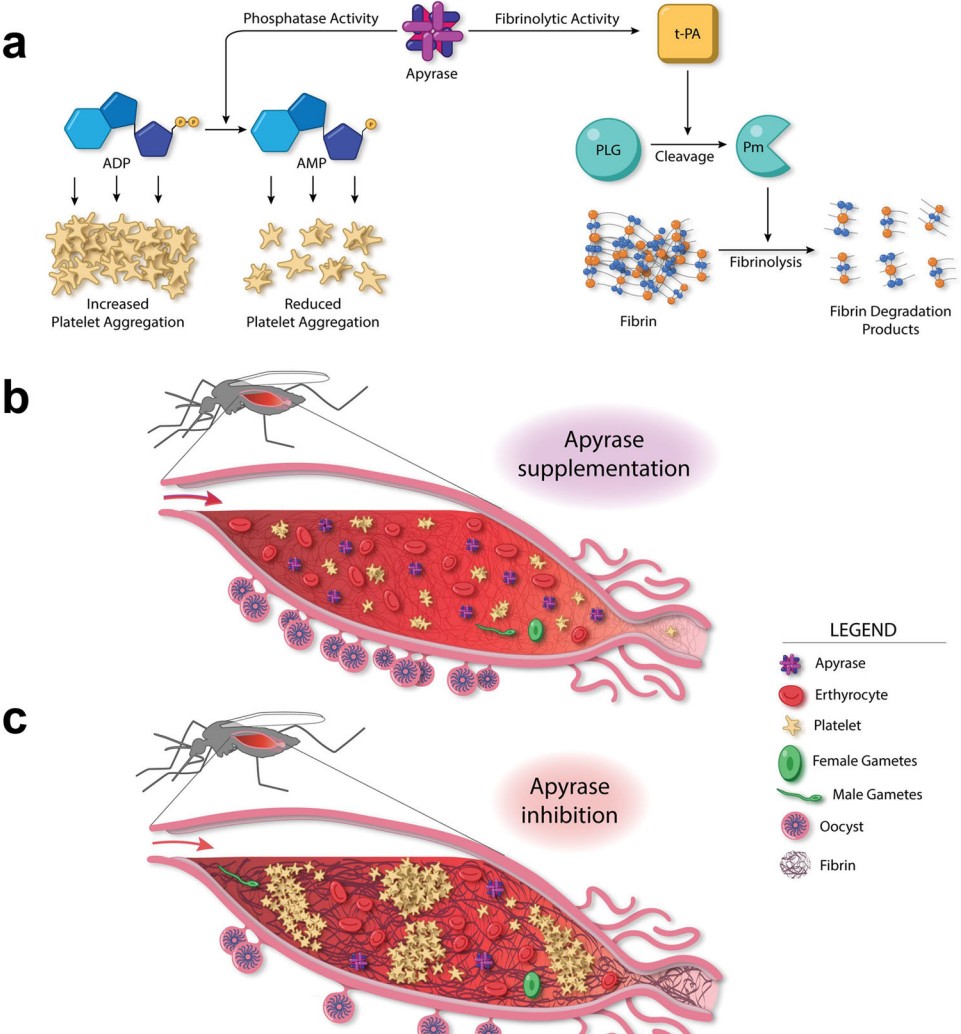

**Fig. 7 | Model for the role of AgApyrase in the mosquito midgut. a** AgApyrase acts as a classical apyrase with phosphatase activity by hydrolyzing ADP to release AMP and phosphate and therefore, prevents ADP-mediated platelet aggregation. AgApyrase activates tissue plasminogen activator (tPA) which in turn activates plasminogen to plasmin. Plasmin enhances the degradation of fibrin. **b** Fibrin polymerization is detected in the mosquito midgut within minutes of an infectious blood meal ingestion. The salivary apyrase ingested during blood feeding, enhances fibrin degradation, and inhibits platelet aggregation thus facilitating the migration of *Plasmodium* gametes in the blood bolus and promoting parasite infection. **c** Inhibition of apyrase with anti-apyrase antibodies results in the formation of a denser fibrin network and increased platelet aggregation, which interferes with *Plasmodium* gamete migration and parasite infectivity.

mosquitoes (4–5 days old) were fed on infected mice once they reached 3–5% parasitemia. After feeding, infected mosquitoes were kept at 19 °C, 80% humidity and 12 h light-dark cycle. To determine the oocyst numbers, midguts were dissected 10 days post-infection, stained with 0.2% mercurochrome, and mature oocysts were counted by light microscopy.

**Salivary gland dissection and saliva collection**
Sugar-fed female adult mosquitoes (4–5 days old) were anesthetized with $CO_2$, transferred to an ice-chilled plate, and their salivary glands were dissected under a stereomicroscope in 1X PBS (137 mM NaCl, 2.7 mM KCl, 4.3 mM $Na_2HPO_4$, and 1.4 mM $KH_2PO_4$, pH 7.4). Salivary gland extract (SGE) was obtained by sonicating the glands on a Branson Sonifier 450 sonicator with the probe immersed for 5 cm in a 100 ml beaker filled with 80 ml ice-cold water. The 1.5 ml tube containing the glands was held in this beaker with the help of forceps such that the base of the tube was under the tip of the probe. Power was set at 6 and three rounds of 50% cycle were run for 1 min. Tissue debris

were cleared by centrifugation at 10,000 g for 10 min at 4 °C. The supernatants were stored at -80 °C until used[51]. Saliva collection was done by placing the back of live mosquitoes on double-sided tape. Mosquito mouthparts (proboscis) were inserted into 10 µl pipette tips containing 10 µl of 1X PBS. Mosquitoes were allowed to salivate for 20 min at room temperature. The saliva was stored at -80 °C until used.

**Single-chain tPA activity assay**
Activation of sc-tPA (100 nM, Innovative Research, Novi, MI, USA, #IHUTPA85SC100UG) was tested in the presence of salivary gland extracts (one, two and four pairs), saliva (10, 20 and 40 µl), or recombinant proteins (100 nM) in 0.01% Tween 20, tris-buffered saline (TBST) (50 mM) Tris-Cl, pH 7.5 and [150 mM NaCl]. The tPA fluorogenic substrate D-Val-Leu-Lys 7-amido-4-methylcoumarin (10 µM) (Millipore-Sigma, Burlington, MA, USA, #A8171) was added, and the change in fluorescence (excitation, 280; emission, 460) was recorded every 1 min over a period of 4 h at 37 °C using the Cytation 5 microplate reader (BioTek Instruments, USA).

## Salivary gland fractionation and mass-spectrometry

Salivary gland extracts obtained by sonication from 400 pairs of salivary glands were subjected to size-exclusion chromatography on a Superdex 200 10/300 GL column (Cytiva, Marlborough, MA, USA, #28-9909-44) and 1 ml fractions were collected. Each fraction was tested in the sc-tPA activity assay and the fractions showing activity along with five different fractions showing no activity were analyzed by mass spectrometry at the Protein Chemistry Section of the Research and Technology Branch (NIAID, NIH). Proteins in the samples were reduced, alkylated, digested using trypsin, and cleaned up with sold-phase extraction tips (OMIX10, Agilent). The peptide samples were loaded to a PepMap 100 C18 trap column (Thermo Scientific, 75 μm diameter, 2 cm length, 3 μm particles), and separated using a PepMap C18 analytical column (Thermo Scientific, 75 μm diameter, 25 cm length, 2 μm particles) using a 2-h acetonitrile gradient (0 – 40% in 100 min, 40–80% in 5 min, hold at 80% for 5 min, 80 – 0 % in 5 min, and hold at 0% for 5 min) with the flow rate of 300 nL/min using EASY nLC 1000 (Thermo Scientific). The MS data acquisition was carried out in a data-dependent acquisition mode using Orbitrap Fusion™ Mass Spectrometer (Thermo Scientific). The survey MS1 scans were recorded every 2 s with the Orbitrap mass analyzer at the 120,000-resolution setting. For the multiple-charged precursors, ddMS2 was performed with the quadrupole isolation of m/z 1.6 window, dissociated by CID, and scanned using the Linear Ion Trap mass analyzer. The dynamic exclusion was utilized for a duration of 30 s. The data were analyzed using PEAKS X plus against the *Anopheles gambiae* protein database from VectorBase.org (VB2019-06) and cRAP commonly contaminating protein database from theGPM.org. All peptides were filtered at a 0.5% FDR, and proteins identifications required a minimum of 2 peptides.

## Cloning, expression, and purification of recombinant candidate proteins

DNA sequences of eight candidate proteins were codon-optimized for mammalian expression systems, synthesized, and cloned in the plasmid VR2001-TOPO (Vical Incorporated, San Diego, CA, USA) by BioBasic Inc. The plasmids harboring the desired candidate genes were transformed in One Shot TOP10 Chemically Competent *E. coli* (Invitrogen, Waltham, MA, USA, # C404010). Plasmid DNA was purified using NucleoBond PC 2000 plasmid megaprep kit (Macherey-Nagel, #740549). Recombinant protein expression was carried out at the Protein Expression Laboratory, Advanced Technology Research Facility (NCI, Frederick, MD, USA) Briefly, FreeStyle 293 F cells (Thermo Fisher Scientific, Waltham, MA, USA, #R79007) were transfected with 1 mg of plasmid DNA, and supernatants were collected after 72 h of transfection. His-tagged recombinant proteins were purified from the supernatants by affinity chromatography followed by size-exclusion chromatography, using Nickel-charged HiTrap Chelating HP (Cytiva, Marlborough, MA, USA, #17040903) and Superdex 200 10/300 GL columns, respectively (Cytiva, Marlborough, MA, USA, #28-9909-44). All protein purifications were carried out using the AKTA pure system (Cytiva, Marlborough, MA, USA). Purified proteins were resolved in a NuPAGE Novex 4–12% Bis-Tris protein gels (Thermo Fisher Scientific, Waltham, MA, USA, # NP0321BOX) and visualized by Coomassie blue stain using the eStain protein stain system (GenScript, Piscataway, NJ, USA). Protein identity was verified by Edman degradation by the Protein Chemistry Section of the Research Technologies Branch, NIAID, NIH.

## AgApyrase interaction with sc-tPA

To analyze the interaction of AgApyrase with sc-tPA, 96 wells flat-bottom clear plates (Costar, Corning, NY, USA, #COS3915) were coated overnight at 4 °C with 20 nM rAgApyrase and an unrelated protein amongst the eight candidate proteins (AGAP007393) in carbonate bicarbonate buffer pH 9.6 (Millipore-Sigma, Burlington, MA, USA, #SRE0102). Plates were washed with 0.05% v/v Tween−20 in TBS

(25 mM Tris, 150 mM NaCl, pH 7.4) between all steps before using the stop solution (2 N H$_2$SO$_4$). Plates were blocked with blocking buffer (TBS-Tween with 5% Skim milk) for 1 h at 37 °C. After the blocking step, plates were coated with sc-tPA (20 nM) and incubated at 37 °C for 2 h. Anti-sc-tPA (Innovative Research, Novi, MI, USA, #ISHAHUTPAA-P100UG) at 1:1500 dilution was used as a primary antibody to detect protein-protein interactions. Primary antibodies were incubated for 1 h at 37 °C. Donkey anti-sheep IgG labeled with horse radish peroxidase (Abcam, Waltham, MA, USA), #ab6900 (1:2500) in blocking buffer was added to the wells and the plates were incubated for 1 h at 37 °C. Plates were developed with 1-Step ultra TMB-ELISA substrate solution (Thermo Fisher Scientific, Waltham, MA, USA, #34028) and incubated at room temperature for 15–30 min. 2 N H$_2$SO$_4$ solution was added to stop the reaction and absorbance was measured at 450 nm using Cytation 5 microplate reader (BioTek Instruments, USA).

## Plasmin activity assay

Plasmin activity was measured using the specific plasmin chromogenic substrate D-Val-Leu-Lys 4-nitroanilide dihydrochloride (Millipore-Sigma, Burlington, MA, USA, #V0882). All proteins were prepared in phosphate-lysine buffer (10 mM potassium phosphate, 70 mM sodium phosphate, 100 mM lysine, pH 7.5). Phosphate-lysine buffer, rAgApyrase (100 nM), sc-tPA (100 nM), tc-tPA (100 nM), plasminogen (400 nM), and rAgApyrase with plasminogen were used as negative controls. Plasmin (400 nM), sc-tPA and tc-tPA with plasminogen were used as positive controls. The respective samples were added to a 96-well flat bottom clear plate and 3.58 mg/mL of chromogenic substrate were added. Plasmin activity was monitored by continuously measuring the rate of change of absorbance at 405 nm for 20 min using Cytation 5 microplate reader (BioTek Instruments, USA).

## AMP hydrolysis activity assay

A malachite green assay kit (R&D Systems, Minneapolis, MN, USA, #DY996) was used to determine the phosphate released by rAg5'NTE using AMP (Millipore-Sigma, Burlington, MA, USA, #A1752) as a substrate. The manufacturer's microplate assay protocol was followed. Briefly, phosphate standards from 0 μM to 100 μM were prepared and 50 μl of CD73 (0.1 μg) (Millipore-Sigma, Burlington, MA, USA, # N1665) was used as a positive control. 10 μl malachite green reagent A was added, and the plate was incubated for 10 min at room temperature. 10 μl of malachite green reagent B was added and the plate was incubated for 20 min at room temperature. The absorbance was measured at 620 nm using Cytation 5 microplate reader.

## ATP and ADP hydrolysis activity assay

Apyrase activity was determined by using a micro-colorimetric method for measuring release of inorganic phosphate from ATP or ADP as previously described[52,53]. Briefly, a buffer containing 50 mM Tris pH 8.3, 150 mM NaCl and 5 mM CaCl$_2$, was mixed with potato apyrase (positive control) (Millipore-Sigma, Burlington, MA, USA, #A6410), buffer alone (negative control) and rAgApyrase in the wells of a 96-well microtiter plate with a final volume of 0.1 ml containing 2 mM of either ATP (Millipore-Sigma, Burlington, MA, USA, #A26209) or ADP (Millipore-Sigma, Burlington, MA, USA, #A2754). This mixture was incubated at 37 °C for 10 min. The reaction was stopped by the addition of 28 μl of a stop reaction mixture: 3 μl of the reducing reagent (mixture of 0.2 g 1-amino-2-naphtol-4-sulfonic acid (Millipore-Sigma, Burlington, MA, USA, #08751) with 0.2 g sodium bisulfite (Millipore-Sigma, Burlington, MA, USA, #243973) and 1.2 g sodium sulfite (Millipore-Sigma, Burlington, MA, USA, #239321) diluted to give a final concentration of 25 mg/ml) and 25 μl of 1.25% ammonium molybdate (Millipore-Sigma, Burlington, MA, USA, # 277908) in 2.5 N H$_2$SO$_4$[52]. The absorbance was measured at 650 nm using Cytation 5 microplate reader. The concentration of phosphate was determined by interpolation with a phosphate standard curve.

## Western blot assay

Western blot assay was used (i) to study the interaction of sc-tPA and AgApyrase and (ii) determine the specificity of the polyclonal anti-apyrase antibodies. To study the interaction of sc-tPA with AgApyrase, sc-tPA at different concentrations (200 nM, 500 nM and 1 μM) was incubated with 500 nM rAgApyrase at 37 °C for 4 h. sc-tPA and tc-tPA were used as positive controls. To determine the specificity of the polyclonal anti-rAgApyrase antibodies, human serum (2 μl) and human plasma (2 μl) along with human blood-fed mosquitoes and unfed mosquitoes were tested with anti-apyrase antibodies. The mosquito midguts were dissected 30 min post blood feeding in PBS supplemented with 5 mM EDTA, 0.1 mM phenylmethylsulfonyl fluoride (Millipore-Sigma, Marlborough, MA, USA), and protease inhibitors (Millipore-Sigma, Marlborough, MA, USA). A total of five midguts were pooled in 50 μl of PBS, then the midguts were macerated on ice. For both the experiments, reducing SDS–polyacrylamide gel electrophoresis (PAGE) sample buffer (LI-COR Biotechnology, Lincoln, NB, USA) was added to the samples and were separated by SDS-PAGE and transferred to polyvinylidene difluoride membranes (Millipore-Sigma, Marlborough, MA, USA). The membrane was blocked with Intercept® (PBS) Blocking Buffer (LI-COR Biotechnology, Lincoln, NB, USA) overnight at 4 °C following incubation with mouse anti-rAgApyrase antibody (1:10,000 in blocking buffer) or rabbit anti-tPA antibodies (1:2500) (Millipore-Sigma, Marlborough, MA, USA). Membranes were incubated with anti-mouse or anti-rabbit horseradish peroxidase (1:2500 in blocking buffer) for 1 h at room temperature and detected using SuperSignal Western Dura PLUS Chemiluminescent Substrate (Thermo Fisher Scientific).

## Multiple sequence alignment, phylogenetic analysis, and structure prediction

Multiple sequence alignment was done for AgApyrase (gene ID: AGAP011026) with the 5′ NTE sequences available from different mosquito species and hematophagous arthropods using CLUSTAL Omega[54]. The sequences used were *Anopheles stephensi* (ASTE008450); *Anopheles darlingi* (ADAC007226); *Culex quinquefasciatus* (CPIJ019168) *Aedes aegypti* (AAEL006347); *Aedes albopictus* (AALF004988); *Ixodes scapularis* (ISC1N003760); *Glossina morsitans* (GMOY012313); *Rattus norvegicus* (AAH81806); *Culicoides sonorensis* (CSON007783); *Lutzomyia longipalpis* (AAD33513); *Phlebotomus duboscqi* (ABI20147); *Cimex lectularius* (AAD09177); *Drosophila melanogaster* (CAL26011); *Xenopus tropicalis* (NP988940); *Gallus gallus* (NP001026752); *Homo sapiens* (NP620148); *Mus musculus* (NP083778). To understand the evolutionary position of Ag5′NTE, a phylogenetic tree was constructed using MEGA 6.0[55]. The evolutionary history was inferred using the maximum-likelihood method[56]. The bootstrap consensus tree inferred from 1000 replicates was taken to represent the evolutionary history of the taxa analyzed[57] where the analysis involved 18 amino acid sequences. All positions containing gaps and missing data were eliminated. To obtain the phylogenetic tree, neighbor-join and BioNJ algorithms were applied to a matrix of pairwise distances estimated using a JTT model, and then the topology with superior log likelihood value was selected[55].

Models of AgApyrase-tPA complex were generated using RoseTTAFold CC[58], OmegaFold[59], and AlphaFold2 Multimer[60] using default parameters. Only the AlphaFold2 models resulted in compact arrangements with OmegaFold suggesting some contacts between the two molecules. The models were subjected to further optimization using Yasara[61] version 22.9.24 and the hm_build.mcr homology modeling macro using the AlphaFold and OmegaFold models as templates and default parameters. This procedure improved the model's overall quality resulting in a final arrangement with a Z-score of -0.914 with both the Apyrase and tPA models in good agreement with known structures. The initial and final models, macros, and comparisons against known structures were deposited in Zenodo with DOI 10.5281/

zenodo.11151002. Several sequences of *Anopheles* Apyrases were aligned using Clustal Omega[54] as implemented in JalView version 2.11.2.6[62].

## Biolayer interferometry (BLI)

We assessed the binding of sc-tPA to rAgApyrase using an Octet R8 instrument (Sartorius). The experiments were conducted in 10x KB buffer (1xPBS pH 7.4 containing 0.1% BSA, 0.2% Tween 20) with agitation set to 1000 rpm. The BLI measurements were done in solid black 96-well plates (Geiger Bio-One) at 30 °C. rAgApyrase was biotinylated with EZ-Link™ Sulfo NHS-LC-LC-Biotin (Thermo Fisher Scientific) as suggested by the manufacturer's protocol. Streptavidin (SA) biosensors were loaded with biotinylated rAgApyrase from 4.7 μg/ml solution and equilibrated in the buffer. The sensors were then exposed to 250-1000 nM sc-tPA to probe the binding and placed in the buffer for 300 s to follow sc-tPA dissociation. The data were analyzed and fitted using Octet® BLI Analysis 12.2.13 software.

## Ex vivo platelet-aggregation assay

Platelet aggregation assay was performed as described by Martin-Martin et al., 2020[63]. Platelet rich plasma was obtained from normal healthy donors enrolled in a protocol approved by the National Institutes of Health Clinical Center Institutional Review Board (NIH protocol 99-CC-0168 "Collection and Distribution of Blood Components from Healthy Donors for In Vitro Use"). Blood donors provided written informed consent, and platelets were de-identified prior to distribution. Platelet aggregation was measured using a light transmission aggregometer (Chrono-Log Corporation, Havertown, PA, USA). Briefly, 300 μL of platelet rich plasma, diluted to ~2.5 × 10^5 platelets/μL in HEPES-Tyrode's buffer (137 mM NaCl, 27 mM KCl, 12 mM NaHCO₃, 0.34 mM sodium phosphate monobasic, 1 mM MgCl₂, 2.9 mM KCl, 5 mM HEPES, 5 mM glucose, 1% BSA, 0.03 mM EDTA, pH 7.4) were pre-stirred in the aggregometer for 1 min to monitor pre-aggregation effects. Recombinant AgApyrase (3 μM) or Tyrode's buffer (negative control) were added to the platelet rich plasma and were placed in a Chrono-Log aggregometer model 700 (Chrono-Log Corporation) and stirred at 1200 rpm at 37 °C for 1 min prior to the addition of ADP (0.3 μM, Chrono-Log Corporation) as an agonist.

For the whole blood platelet aggregation assay, whole blood in acid citrate dextrose was obtained from healthy donors enrolled in a protocol approved by the National Institutes of Health Clinical Center Institutional Review Board (NIH protocol 99-CC-0168 "Collection and Distribution of Blood Components from Healthy Donors for In Vitro Use"). Whole blood was diluted 1:1 (v/v) in saline buffer (0.9%) at pH 7.4 and pre-stirred for 5 min at 37 °C. ADP (Chrono-Log Corporation) was used as agonist at a final concentration of 10 μM in 950 μL of diluted whole blood. AgApyrase (2 μM) was mixed with ADP (10 μM) for 3 min before being added to the blood sample. Samples were analyzed in a Chrono-Log aggregometer model 700 (Chrono-Log Corporation). Technical duplicates were performed for each experimental condition, and the data were plotted with GraphPad Prism (v. 9.3.1).

## Anti-AgApyrase polyclonal antibody generation

BALB/c mice were inoculated with 50 μg of recombinant AgApyrase protein in Magic Mouse adjuvant (Creative Diagnostics, Shirley, MA, USA, #CDN-A001). As control, were inoculated with adjuvant only. Mice were boosted 3 weeks after the primary injection with the same amount of protein. Terminal bleeds were performed at 35 days post first immunization. All injections and terminal bleeds were carried out by the NIAID/NIH animal facility personnel. The antibody levels were determined by coating recombinant AgApyrase (1 μg/ml) on a 96-well, flat bottom clear plate (Costar, Corning, NY, USA) overnight at 4 °C. Wells were washed three times in TBS-Tween and blocked with blocking buffer (5% skim milk in 1XTBST). After 1 h, wells were washed three times in TBS-Tween incubated with adjuvant control or

AgApyrase serum samples (1:25,000 dilution) in blocking buffer for 1 h. Wells were washed three times and incubated with HRP-labeled anti-mouse IgG H + L (1:2500 dilution in blocking buffer) (Cell Signaling Technology, Danvers, MA, USA, #7076P2). Plates were developed with 1-Step ultra TMB-ELISA substrate solution and incubated at room temperature for 15–30 min. 2 N $H_2SO_4$ solution was added to stop the reaction and absorbance was measured at 450 nm using Cytation 5 microplate reader.

## Immunohistochemistry of mosquito midguts

Immunohistochemistry was performed on either whole mosquitoes or dissected midguts to determine the ingestion of salivary apyrase by the mosquito. Unfed whole mosquitoes (2 days old, negative control) and blood-fed mosquito midguts (human blood NIH protocol 99-CC-0168 "Collection and Distribution of Blood Components from Healthy Donors for In Vitro Use") were dissected 30 min post-feeding. The mosquitoes and mosquito midguts were fixed in 10% neutral buffered formalin, processed with Leica ASP6025 tissue processor (Leica Microsystems), embedded in paraffin and sectioned at 5 μm for histological analysis. These formalin-fixed paraffin-embedded tissues sections were used to perform immunohistochemical staining using mouse anti-apyrase antibody (1:10,000) and rabbit anti-AAPP antibody (1:50) to stain unfed and blood-fed mosquitoes. Antibody diluent (Agilent, Santa Clara, CA, USA, # S302283-2) was used as a negative reagent control to replace the primary antibodies. Staining was carried out on the Bond RX (Leica Biosystems) platform according to manufacturer-supplied protocols. Briefly, 5 μm-thick sections were deparaffinized and rehydrated. Heat-induced epitope retrieval (HIER) was performed using Epitope Retrieval Solution 1, pH 6.0, heated to 100 °C for 20 min. The specimen was then incubated with hydrogen peroxide to quench endogenous peroxidase activity prior to applying the primary antibody. Detection with DAB chromogen was completed using the Bond Polymer Refine Detection kit (Leica Biosystems, #DS9800). Slides were finally cleared through gradient alcohol and xylene washes prior to mounting and placing cover slips. Sections were examined by a boarded-certified veterinary pathologist using an Olympus BX51 light microscope and photomicrographs were taken using an Olympus DP73 camera.

To determine platelet aggregation inside the mosquito midgut, a group of mosquitoes (labeled as "Before" in the figures) were fed on Swiss-webster mice for 15 min. The same mouse was IV injected with 200 μg of rAgApyrase and a second group of mosquitoes (labeled as "After" in the figures) were fed on it for 15 min. Mosquito midguts were dissected 30 min post-feeding and fixed in 10% neutral buffered formalin and were processed as described above. A rabbit polyclonal CD62p (Bioss, Woburn, MA, USA, #Bs-0561R), which is a marker for platelet activation and aggregation was used to stain the formalin-fixed paraffin-embedded tissues sections with dilution of 1:200. The sections were then processed for detection as described above.

To quantify the amount of fibrin degradation in the mosquito midgut, the above mosquito midgut samples were stained with rabbit anti-fibrinogen antibody (1:200; Abcam #227063). The sections were then processed for detection as described above.

## Quantitative PCR

Midguts and salivary glands were harvested from *An. gambiae* mosquitoes, either sugar-fed or blood-fed. Blood-fed mosquitoes were fed on Balb/C mice; midguts were dissected 24 h post-feeding in phosphate-buffered saline (PBS). Salivary glands were extracted from 4 day-old sugar-fed females. Midguts (15 per pool) and salivary glands (15 pairs) were homogenized in 1 mL TRIzol LS Reagent (Thermo Fisher Scientific). RNA was isolated according to the manufacturer's instructions, with specific modifications for salivary glands: RNA precipitation was performed using isopropyl alcohol (1:1 ratio), and 20 μL of linear acrylamide (Thermo Fisher Scientific) was added to the precipitation

mix to enhance precipitation. Samples were mixed by inversion (10 times), incubated at room temperature for 10 min, and centrifuged at 12,000 RCF for 10 min at 4 °C. RNA pellets were washed twice with 75% ethanol, using at least 1 mL ethanol per 1 mL TRIzol. The RNA was then dried and rehydrated in 30 μL of RNase-free water, gently mixed, and heated at 55 °C for 10 min for complete dissolution. Total RNA was prepared in nuclease-free water; 1 μg from midguts was used for cDNA synthesis with the Quantitect Reverse Transcription Kit (Qiagen, Germantown, MD, USA). Salivary gland RNA, typically lower in yield, was prepared in 18 μL of water and directly used for cDNA synthesis. Quantitative PCR (qPCR) was performed to assess Apyrase (AGAP011026) gene expression, using the PowerUp™ SYBR™ qPCR Kit (Thermo Fisher Scientific, A25742) with specific primers (F- TGC CATCGATCACTCCTTCA, R- CGATGCTCTGTACACGTTCG). A 119-bp fragment was amplified, and expression levels were normalized against *Anopheles gambiae* ribosomal protein S7 (RpS7, AGAP010592) using ΔΔCt method (Livak and Schmittgen, 2001; Pfaffl, 2001). RpS7 primers were F-AGAACCAGCAGACCATC and R-GCTGCAAACTTCGGCTATTC. Statistical analyses were performed using an unpaired *T*-test (Graph-Pad, San Diego, CA, USA). Each experiment included three biological replicates per condition for midguts and six for salivary gland extracts.

## Scanning electron microscopy

To determine the effect of AgApyrase on fibrin polymerization in the mosquito midgut, a group of mosquitoes (labeled as "Before" in the figures) were fed on Swiss-webster mice for 15 min. The same mouse was IV injected with 200 μg of rAgApyrase and a second group of mosquitoes (labeled as "After" in the figures) were fed on it for 15 min. Mosquito midguts were dissected 30 min and 4 h post-feeding. Mosquito midguts were fixed in 2.5% glutaraldehyde / 4% paraformaldehyde in 0.1 M sodium cacodylate buffer. The samples were rinsed in 0.1 M sodium cacodylate buffer three times for 5 min each and post-fixed with 0.5% $OsO_4$ / 0.8% $K_4Fe(CN)_6$ in 0.1 M cacodylate buffer for 1 h. Following three more 5 min rinses in 0.1 M cacodylate buffer the midguts were dehydrated in a graded ethanol series. The midguts were dropped in liquid nitrogen and cut in half with a razorblade. The midgut halves were placed back into 100% ethanol and dried in a Baltec CPD030 critical point drier (Balzers). The samples were mounted on aluminum SEM stubs using carbon sticky tape (Electron Microscopy Sciences, Hatfield, PA, USA) and coated with 12 nm of iridium in an EMS300T D sputter coater (Electron Microscopy Sciences) before being viewed at 2.0 kV in a Hitachi SU8000 field emission scanning electron microscope (Hitachi, High-Tech in America) in secondary electron mode. The same protocol was also used for midguts from mosquitoes fed on adjuvant control and rAgApyrase immunized mice. Midguts from mosquitoes feeding on immunized mice were dissected 30 min post-feeding.

## Transmission electron microscopy

*An. gambiae* mosquito midguts obtained from "Before" and "After" groups discussed above were fixed in 4% paraformaldehyde in 0.1 M phosphate buffer. Midguts were then rinsed in 0.1 M cacodylate buffer 3 times for 5 min at room temperature each before being dehydrated in a graded ethanol series from 50% to 90% ethanol. The samples were then infiltrated in a 1:1 mixture of 90% ethanol and LR White resin with accelerator (Electron Microscopy Sciences, Hatfield, PA, USA) for 16 h at 4 °C on a shaker. After infiltrating the tissue in 100% LR White resin for 8 h, the midguts were embedded in fresh LR White resin in gelatin capsules (Electron Microscopy Sciences) at 50 °C under vacuum for 48 h. Thin sections were cut on a Leica UC6 ultramicrotome (Leica Microsystems, Wetzlar Germany) and collected on 200 mesh formvar/carbon coated nickel grids (Electron Microscopy Sciences). Immunolabeling was done using the drop flotation method where grids are floated on 10 – 50 μL droplets of the solution for each step. Grids were blocked with 1% BSA / 1% gelatin / 0.01% Tween20 in PBS for 10 min

then labeled with primary antibody (anti-fibrinogen Millipore--Sigma, Marlborough, MA, USA #341552 or anti-TER-119, BD Biosciences, San Jose, CA, USA #550565) diluted 1:25 in blocking buffer for 1 h. The grids were washed twice with blocking buffer before labeling with secondary antibodies (goat anti-rat or goat anti-rabbit, 10 nm, Electron Microscopy Sciences), rinsed three times in 1X PBS, and rinsed three additional times in filtered dH₂O. The grids were stained with 1% uranyl acetate for 8 min and imaged in a Hitachi 7800 transmission electron microscope at 80 kV (Hitachi High-Tech in America).

### D-Dimer assay

To measure D-dimer formation in mosquito blood boluses, mosquito midguts from "Before" and "After" groups mentioned above were used. ELISA (from LifeSpan BioSciences), Mouse Fibrin Degradation Product D-dimer ELISA Kit (LS Bio, #LS-F6179) was used to test the level of D-dimer in 5 and 10 mosquito midguts. The mosquito midguts were dissected in 1X PBS 30 min post feeding, homogenized thoroughly, and subjected to three cycles of freeze (-20 °C)/ thaw (room temperature). The cell debris were removed by centrifugation at 5000 g for 5 min at 4 °C and the supernatant was used for the assay. The assay was performed as per manufacturer's protocol. Briefly, standard solutions from a range of 0 – 50,000 pg/ml were prepared for a standard curve to which 50 µl of detection reagent A were added and the plate was incubated for 1 h at 37 °C, followed by three washes with the wash buffer. 100 µl of detection reagent B were added to the plate and incubated for 30 min at 37 °C. The plate was washed five times with the wash buffer, 90 µl of TMB substrate solution was added to the plate and incubated for 15 min at 37 °C. The reaction was stopped by adding 50 µl of stop solution and the absorbance was recorded at 450 nm using Cytation 5 microplate reader (BioTek Instruments, USA).

### In vitro NET formation assay

Peripheral blood was collected in heparinized tubes after venipuncture. Blood was fractionated via density gradient centrifugation using Ficoll-Paque Plus (GE Healthcare). Neutrophils were isolated by dextran sedimentation, washed with PBS 1X, and red blood cells were lysed with hypotonic salt solution. Neutrophils were resuspended in RPMI serum-free at a density of $1 \times 10^6$ cells/mL. For isolation of platelet rich plasma, peripheral blood was collected in heparinized tubes, blood was centrifuged at 113 g for 10 min, then supernatant plasma containing platelets was transfer to a new sterile tube and centrifuged at 1016 g for 10 min at room temperature. After centrifugation, 2/3 parts of volume plasma were discarded, and platelets were suspended in a minimum quantity of plasma (2–4 mL) and counted. Neutrophils were preincubated with 1–5 uM of the mosquito rAgApyrase. After preincubation, neutrophils were incubated with $2 \times 10^5$ platelets for 2–3 h at 37 C. Cells were fixed with 4% PFA and stored at 4 °C until immunofluorescence analysis was performed.

Fixed neutrophils were washed with PBS and blocked with 0.2% porcine gelatin (Sigma) for 1 h at room temperature. Cells were incubated with primary Ab anti-MPO (Dako); for 30 min in a humid chamber at 37 °C. Coverslips were washed three times with PBS and incubated with secondary antibody (Goat anti-rabbit IgG, Invitrogen) for 30 min at 37 °C. Nuclei were counterstained with Hoechst for 10 min at RT. After three washes, coverslips were mounted on glass slides using Prolong Gold solution (Invitrogen). Slides were visualized in a Zoe microscope. Three pictures of each experiment were taken, and NETs were counted.

### *Plasmodium berghei* infection

To determine the effect of AgApyrase on *Plasmodium* infection, *P. berghei* infections were performed using a transgenic line expressing mcherry and luciferase[50] and maintained by serial passages in female Swiss webster mice. Parasitemia was assessed by light microscopy by using methanol-fixed blood-smears from infected mice and stained with 10% Giemsa. A group of mosquitoes (Before) were fed on infected Swiss-webster mice for 15 min. The same mouse was IV injected with rAgApyrase (200 µg) and a second group of mosquitoes (After) were fed on it for 15 min. This assured that the blood-meal in both the groups had similar parasitemia. For heat-denatured protein assay, the protein was incubated at 65 °C for 30 min and then IV injected. Infected mosquitoes were kept at 19 °C, 80% humidity and 12 h light-dark cycle. To determine oocyst numbers, midguts were dissected 10 days post-infection, stained with 0.2% mercurochrome, and mature oocysts were counted by light microscopy. The same protocol was followed for mosquitoes fed on adjuvant control and rAgApyrase immunized mice.

### Sporozoite transmission assay

For mosquito challenge: Adjuvant control and *Ag*Apyrase immunized mice were challenged with the bite of five *An. stephensi* mosquitoes infected with *P. berghei* sporozoites expressing mCherry and luciferase. The infection status of mosquito salivary glands was verified before the challenge by the presence of strong red fluorescence in the thoracic cavity.

For intravenous sporozoite challenge: Adjuvant control and rAgApyrase immunized mice were anesthetized by peritoneal injection of ketamine (50 mg/kg body weight) and xylazine-hydrochloride (10 mg/kg body weight), and then 5,000 *P. berghei* sporozoites expressing mCherry and luciferase, and isolated from salivary glands, were injected intravenously in the lateral tail veins in a total volume of 200 µl of PBS.

Parasite liver load was determined by measuring luciferase activity in the mouse liver as previously reported[46]. Forty-two h after sporozoite injection or mosquito-bite challenge, the mouse abdominal hair was removed using Nair cream, the challenged mice were intraperitoneally injected with 100 µL of D-luciferin (30 mg/mL) (Perkin Elmer, Waltham, MA, USA, #122799), and then anesthetized in an isoflurane chamber. After mice were immobilized, they were placed in an IVIS Spectrum imager to evaluate bioluminescence by measuring the radiance from the abdomen for 5 min. Mice were returned to their cage to monitor parasite development in the blood by the pre-patency assay. Infection patency was monitored by Giemsa-stained blood smears for 12 days after infection. A single experiment included five mice for each treatment.

### Statistics and reproducibility

Assays for the activation of tPA and neutrophil activation were analyzed using One-way ANOVA with Šídák's multiple comparisons test. Nucleotidase activity assays, D-dimer formation, were analyzed by two-tailed unpaired *t*-test. tPA interaction with salivary candidate activators was analyzed by One-way ANOVA with Dunnet's multiple comparison test. IHC quantification and transmission blocking assays were analyzed by Mann Whitney test. Statistical significance analyses were performed using GraphPad Prism version 9.0. No statistical method was used to predetermine sample size. No data were excluded from the analyses. The experiments were not randomized and the Investigators were not blinded to allocation during experiments and outcome assessment.

### Reporting summary

Further information on research design is available in the Nature Portfolio Reporting Summary linked to this article.

## Data availability

All data needed to evaluate the conclusions are included in the paper, the Supplementary Materials have been deposited in online repositories. The mass spectrometry proteomics data have been deposited to the ProteomeXchange Consortium via the PRIDE partner repository with the dataset identifier PXD052155 and 10.6019/PXD052155. Source data are provided with this paper.

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

## Acknowledgements

The authors are grateful to Jose Ribeiro and David Garboczi for critical review of the manuscript, Andre Laughinghouse and Kevin Lee for insectary support; Allison Booth and DeAndre Bacon for technical support with animals; L. Renee Olano, Glenn Nardone and Motoshi Suzuki for support with mass spectrometry analysis. This work was funded by NIH Distinguished Scholars Program and the Intramural Research Program of the Division of Intramural Research (AI001250-01), National Institute of Allergy and Infectious Diseases (NIAID), NIH to J.V.R.; Malaria Research Program Fellowship to Z.R.P.

## Author contributions

Conceptualization: Z.R.P., T.L.A.S., J.V.R. Methodology: Z.R.P., T.L.A.S., J.V.R.; Investigation: Z.R.P., M.M., B.C., E.P.M., C.C.R., P.C.L-V., I.M-M., Y.F-G., R.E.C., L.M., A.G.G., N.S., I.M.M., D.A.A., D.N.G., E.F., M.J.K., E.C., J.V.R.; Formal analysis: Z.R.P., and J.V.R.; Writing—original draft: Z.R.P. and J.V.R.; Supervision: J.V.R.

## Funding

## Competing interests

The authors declare no competing interest.

## Additional information

[1]Laboratory of Malaria and Vector Research, National Institute of Allergy and Infectious Diseases, National Institutes of Health, Rockville, MD 20852, USA. [2]Infectious Disease Pathogenesis Section, National Institute of Allergy and Infectious Diseases, National Institutes of Health, Rockville, MD 20852, USA. [3]Microscopy Unit, Research Technologies Branch, Rocky Mountain Laboratories, National Institute of Allergy and Infectious Diseases, National Institutes of Health, Hamilton, MT 59840, USA. [4]Systemic Autoimmunity Branch, National Institute of Arthritis and Musculoskeletal and Skin Diseases, National Institutes of Health, Bethesda, MD 20892, USA. [5]Department of Molecular Microbiology and Immunology, Malaria Research Institute, Johns Hopkins Bloomberg School of Public Health, Baltimore, MD, USA. [6]Integrated Data Science Section, Research Technologies Branch, National Institute of Allergy and Infectious Diseases, National Institutes of Health, Bethesda, MD 20892, USA. [7]Structural Biology Section, Research Technologies Branch, National Institute of Allergy and Infectious Diseases, National Institutes of Health, Bethesda, MD 20892, USA. [8]Present address: Laboratory of Medical Entomology, National Center for Microbiology, Instituto de Salud Carlos III, 28220 Majadahonda, Madrid, Spain. ✉e-mail: joel.vega-rodriguez@nih.gov

