## [Peer Review File · Nature Communications]

Mosquito salivary apyrase regulates blood meal hemostasis and facilitates malaria parasite transmissionREVIEWER COMMENTS

Reviewer #1 (Remarks to the Author):

This article is well-written and the discovery is significant to communities. The finding also suggests the great potential for malaria control application. The methods are reliable and the results are solid.

A couple of minor errors are listed below:

- 1) Figure 2F, two labels (AgAPy+sc-tPA) are the same. Please correct them. These should be described in text (line 159-162).
- 2) Figure 2I, The tc-tPA (two chain tPA) showed one band, while sc-tPA showed two bands. Please clarify the definition and difference of tc-tPA and sc-tPA.
- 3) In discussion, authors talked about the limitation of *P. berghei* infection system. It would be very helpful to show whether mosquito salivary apyrase also inhibit human blood coagulation in vitro, a key experiment for applying this discovery for malaria control in the future.

Reviewer #2 (Remarks to the Author):

Here, the authors identify and characterize a novel protein from mosquito saliva and show that it impacts transmission of the malaria parasite *Plasmodium*, both from host-to-mosquito and from mosquito-to-host. The authors show that the protein, termed AgApyrase, is capable of initiating an enzymatic cascade that reduces blood clotting. In a series of well-designed experiments, the authors isolate and identify the protein from mosquito saliva, then express it recombinantly and verify its anti-clotting activity through a variety of in vitro and in vivo assays. Based on previous observations that blood viscosity affects the ability of *Plasmodium* to successfully infect mosquitoes, the authors demonstrate that the anti-clotting activity of AgApyrase enhances the ability of parasites to invade the mosquito midgut after being taken up in an infected blood meal. The authors also investigate the impact of the protein on transmission of parasites from mosquito to host. Strikingly, the ability of parasites to infect mice via mosquito bite was significantly reduced when the animals were immunized with AgApyrase, implying that the protein mediates parasite infectivity at this stage as well.

The experimental design is sound and the findings will be of interest to those in the field. I recommend publication after minor revisions, detailed below.

Issues

Section beginning on Line 302, "Salivary apyrase drives sporozoite transmission"

In the title "Salivary apyrase drives sporozoite transmission", I suggest replacing "drives" with "facilitates" or "augments". AgApyrase activity is not the driving factor underlying sporozoite transmission, though it clearly helps by way of reducing blood viscosity.

The discussion of the interplay between sporozoites and apyrase needs to be clarified and/or expanded. All of the very thoughtful discussion and work up to this point in the manuscript has very clearly demonstrated how the anti-clotting activity of AgApyrase augments the parasite's ability to invade the mosquito midgut. It is not immediately clear why the same enzyme would be expected to have an impact at a very different stage of the parasite life cycle undergoing a very different set of processes and invading a different tissue in a different organism. The authors briefly allude to the hypothesis that the presence of sporozoites in mosquitoes *decreases* apyrase activity, and that this in turn causes the infected mosquito to bite more often, though it's not immediately clear how these two phenomena are related. The authors demonstrate that mice immunized with AgApyrase, which presumably inhibits AgApyrase activity, are less susceptible to infection by mosquito bite. This seems to contradict the observation from literature that sporozoites have developed a means to decrease apyrase activity. What mechanism would cause the same enzyme that enables invasion of the midgut to enable invasion of the skin? Is it the same anti-clotting activity, or something else?

Reviewer #3 (Remarks to the Author):

The study presents compelling evidence demonstrating the critical role of mosquito apyrase in both blood metabolism processes within the mosquito gut and parasite transmission. Overall, the study is well-executed, but I have some minor comments and suggestions for additional experiments that could enhance the manuscript. Although the paper would benefit from functional in vivo assays directly linking Ag apyrase expression in the salivary glands to the findings, such as through apyrase knockdown or knockout, such experiments are not essential for this study.

In lines 118-119, clarification is needed regarding the selection of fractions A5 and B3 as negative controls among other fractions.

Similarly, in lines 161-162, it would be beneficial to explain why AGAP007393 was chosen among the 8 identified proteins as a control.

For Figure 2H, the methods used to create the structural models should be briefly mentioned in both the text and the legend.

In Figure 2I, annotating the two bands as sc-tPA and tc-tPA instead of unannotated arrows would enhance clarity.

Line 181 should use the past tense for "we postulate" ("we postulated").

Regarding lines 188-189, it may be prudent to avoid claiming to be the first, especially considering prior reports showing ingestion of saliva.

Considering the variability in published transcriptomic studies, it would be strongly advisable for the authors to confirm the absence of apyrase expression in the midgut by real-time RT-PCR (lines 192-193).

Regarding lines 196-205, clarification is needed on whether the authors suggest that the anti-apyrase antibody produces stronger but equally specific signals as the anti-AAPP antibody, or if the polyclonal anti-apyrase antibody reacts with more bloodmeal proteins.

The argument in lines 213-214 regarding observation lacks clarity. Perhaps, it would be better saying: "To determine whether apyrase ingestion increases fibrin degradation ...".

In Figure 3B, the significance of presenting two graphs, one with 5 and the other with 10 midguts, should be explained.

Regarding Figure 3C, the compartmentalization in the fibrinogen signal in non-apyrase-fed midguts, resembling the peritrophic matrix, and its absence in apyrase-fed midguts requires some more explanation, particularly in relation to the fibrin fibre network and the peritrophic matrix. Could apyrase also contribute to the degradation of the peritrophic matrix?

Lines 227-229 and 250-260: Could apyrase act on platelets within the mouse before blood ingestion by the mosquito? Consider conducting blood feeding (either on mouse or human blood) using SMFAs spiked with the apyrase.

In line 275, "both" should replace "either/or."

The argument in lines 303-305 regarding sporozoites regulating apyrase expression to increase the frequency of mosquito bites needs clarification. It would be more intuitive if increased apyrase amounts correlated with successful blood feeding rather than more frequent bites, aligning better with the results on sporozoite infectivity in immunized mice via sporozoite blockade in the skin.

REVIEWER COMMENTS

Reviewer #1 (Remarks to the Author):

This article is well-written and the discovery is significant to communities. The finding also suggests the great potential for malaria control application. The methods are reliable and the results are solid.

A couple of minor errors are listed below:

1) Figure 2F, two labels (AgAPy+sc-tPA) are the same. Please correct them. These should be described in text (line 159-162).

We have corrected the figure label and have added a description in the text.

2) Figure 2I, The tc-tPA (two chain tPA) showed one band, while sc-tPA showed two bands. Please clarify the definition and difference of tc-tPA and sc-tPA.

A clarification has been added on lines 177-178: "sc-tPA is a ~65 kDa protein and it results in smaller ~32 kDa tc-tPA fragments upon activation."

3) In discussion, authors talked about the limitation of *P. berghei* infection system. It would be very helpful to show whether mosquito salivary apyrase also inhibit human blood coagulation in vitro, a key experiment for applying this discovery for malaria control in the future.

We performed this experiment, and the data has been included as Supplementary Figure S9 and elaborated further in the Discussion. AgApyrase does delay blood coagulation *in vitro*, which was measured in an aggregometer.

Reviewer #2 (Remarks to the Author):

Here, the authors identify and characterize a novel protein from mosquito saliva and show that it impacts transmission of the malaria parasite Plasmodium, both from host-to-mosquito and from mosquito-to-host. The authors show that the protein, termed AgApyrase, is capable of initiating an enzymatic cascade that reduces blood clotting. In a series of well-designed experiments, the authors isolate and identify the protein from mosquito saliva, then express it recombinantly and verify its anti-clotting activity through a variety of *in vitro* and *in vivo* assays. Based on previous observations that blood viscosity affects the ability of Plasmodium to successfully infect mosquitoes, the authors demonstrate that the anti-clotting activity of AgApyrase enhances the ability of parasites to invade the mosquito midgut after being taken up in an infected blood meal. The authors also investigate the impact of the protein on transmission of parasites from mosquito to host. Strikingly, the ability of parasites to infect mice via mosquito bite was significantly reduced when the animals were immunized with AgApyrase, implying that the protein mediates parasite infectivity at this stage as well.

The experimental design is sound and the findings will be of interest to those in the field. I recommend publication after minor revisions, detailed below.

Issues:

1. Section beginning on Line 302, "Salivary apyrase drives sporozoite transmission"

In the title "Salivary apyrase drives sporozoite transmission", I suggest replacing "drives" with "facilitates" or "augments". AgApyrase activity is not the driving factor underlying sporozoite transmission, though it clearly helps by way of reducing blood viscosity.

We have changed "drive" for "facilitates".

2. The discussion of the interplay between sporozoites and apyrase needs to be clarified and/or expanded. All of the very thoughtful discussion and work up to this point in the manuscript has very clearly demonstrated how the anti-clotting activity of AgApyrase augments the parasite's ability to invade the mosquito midgut. It is not immediately clear why the same enzyme would be expected to have an impact at a very different stage of the parasite life cycle undergoing a very different set of processes and invading a different tissue in a different organism. The authors briefly allude to the hypothesis that the presence of sporozoites in mosquitoes *decreases* apyrase activity, and that this in turn causes the infected mosquito to bite more often, though it's not

immediately clear how these two phenomena are related. The authors demonstrate that mice immunized with AgApyrase, which presumably inhibits AgApyrase activity, are less susceptible to infection by mosquito bite. This seems to contradict the observation from literature that sporozoites have developed a means to decrease apyrase activity. What mechanism would cause the same enzyme that enables invasion of the midgut to enable invasion of the skin? Is it the same anti-clotting activity, or something else?

We acknowledge the reviewers' valid point regarding the need for clarification on how apyrase facilitates sporozoite transmission. In our previous study (Alves e Silva et al., 2021, PMID: 28690071), we demonstrated that tissue plasminogen activator (tPA) activates plasminogen on the sporozoite surface, and the resultant sporozoite-bound plasmin enhances parasite motility within the dermal extracellular matrix by degrading ECM proteins. As saliva, and consequently apyrase, are co-injected with sporozoites during probing, we propose that apyrase facilitates the activation of tPA, and consequently plasminogen, on the sporozoite surface, thereby promoting parasite motility within the skin. We have expanded the discussion to include this hypothesis, emphasizing the need for further experiments to validate this molecular mechanism in the skin.

The reduction in apyrase activity observed in previous studies can be attributed to a general decrease in salivary proteins within infected salivary glands, as indicated by our analysis of the saliva proteome from infected and uninfected mosquitoes (refer to the unpublished data figure below). Although we observed a slight decrease in apyrase levels in infected glands, this decrease was not statistically significant and may not be biologically relevant. Given that apyrase is the most abundant salivary protein, this reduction does not appear to have significant biological implications. Furthermore, since the increase in probing behavior by infected mosquitoes might not be directly linked to apyrase activity, we have decided to remove this statement from the manuscript. Additionally, we have eliminated discussion regarding apyrase activity, as further research is required to confirm whether this reduction has any impact on mosquito biting behavior and malaria transmission.

Figure 1 : Number of proteins detected in the saliva of uninfected and *Plasmodium*-infected mosquitoes. Saliva was collected by forced salivation and processed for Mass spectrometry. Unpublished data.

Reviewer #3 (Remarks to the Author):

The study presents compelling evidence demonstrating the critical role of mosquito apyrase in both blood metabolism processes within the mosquito gut and parasite transmission. Overall, the study is well-executed, but I have some minor comments and suggestions for additional experiments that could enhance the manuscript.

1. Although the paper would benefit from functional in vivo assays directly linking Ag apyrase expression in the salivary glands to the findings, such as through apyrase knockdown or knockout, such experiments are not essential for this study.

Despite multiple attempts to knock down AgApyrase in *An. gambiae* and *An. stephensi* using RNAi, we were unsuccessful in achieving knockdown at the protein level. One potential explanation is that apyrase is the most abundant protein in mosquito saliva, and upon secretion, it accumulates in the salivary cavities. We are currently collaborating with other laboratories to produce apyrase knockout mosquitoes; however, it remains uncertain whether we will succeed, as this genetic modification may prove lethal to the mosquito.

2. In lines 118-119, clarification is needed regarding the selection of fractions A5 and B3 as negative controls among other fractions.

These fractions were selected because we aimed to choose one fraction preceding and one fraction following our Z series of fractions, which, in turn, showed tPA activation. Another reason was the absence of tPA activation activity in these fractions, as mentioned in the text. Further clarification has been provided in the revised manuscript lines 119-120.

3. Similarly, in lines 161-162, it would be beneficial to explain why AGAP007393 was chosen among the 8 identified proteins as a control.

As shown in figure S1B, after Ag5'NTE, ZPJVR3 (AGAP007393) gave a stronger reading for the slope of fluorescence intensity when compared to all the remaining proteins in consideration. Therefore, we chose this protein as a negative control. A clarification has been added in lines 163-166.

4. For Figure 2H, the methods used to create the structural models should be briefly mentioned in both the text and the legend.

This has been addressed.

5. In Figure 2I, annotating the two bands as sc-tPA and tc-tPA instead of unannotated arrows would enhance clarity.

We have annotated the arrows as requested.

6. Line 181 should use the past tense for "we postulate" ("we postulated").

This has been addressed.

7. Regarding lines 188-189, it may be prudent to avoid claiming to be the first, especially considering prior reports showing ingestion of saliva.

We agree and have removed "is the first to" from the text.

8. Considering the variability in published transcriptomic studies, it would be strongly advisable for the authors to confirm the absence of apyrase expression in the midgut by real-time RT-PCR (lines 192-193).

We have performed qPCR to determine apyrase expression in midguts and salivary glands from sugar-fed mosquitoes and midguts from blood fed mosquitoes. We do not observe expression of apyrase in either of the midguts samples when we compare it to the salivary glands (see graph below). This data has been added to Figure S3D.

9. Regarding lines 196-205, clarification is needed on whether the authors suggest that the anti-apyrase antibody produces stronger but equally specific signals as the anti-

AAPP antibody, or if the polyclonal anti-apyrase antibody reacts with more bloodmeal proteins.

We propose that anti-AgApyrase antibodies recognize more epitopes on the AgApyrase protein compared to anti-AgAAPP antibodies on the AgAAPP protein. As described in the manuscript, the anti-AAPP antibody was developed against a 20-amino acid peptide, resulting in fewer binding sites and, consequently, lower signal intensity on immunohistochemistry compared to the anti-apyrase antibody. The latter was generated against a full-length recombinant protein, enabling recognition of multiple epitopes on the apyrase protein. We have modified the text to clarify this point.

In the original manuscript, we demonstrated that polyclonal anti-apyrase specifically reacts with mosquito apyrase and not with bloodmeal proteins (see Fig S3E of the new manuscript, and Figure 3D in the original submission).

10. The argument in lines 213-214 regarding observation lacks clarity. Perhaps, it would be better saying: "To determine whether apyrase ingestion increases fibrin degradation ...".

This has been addressed.

11. In Figure 3B, the significance of presenting two graphs, one with 5 and the other with 10 midguts, should be explained.

There is no particular significance for including both the graphs in this figure. We observed a significant release of D-dimers with 5 and 10 midguts and decided to include both data in the manuscript.

12. Regarding Figure 3C, the compartmentalization in the fibrinogen signal in non-apyrase-fed midguts, resembling the peritrophic matrix, and its absence in apyrase-fed midguts requires some more explanation, particularly in relation to the fibrin fibre network and the peritrophic matrix. Could apyrase also contribute to the degradation of the peritrophic matrix?

We respectfully disagree with the assertion that the observed compartmentalization resembles the peritrophic matrix, although we acknowledge the presence of some compartmentalization in certain midguts from both experimental groups (before and after AgApyrase supplementation). The peritrophic matrix is typically located in the periphery of the blood meal, which is distinct from the compartmentalization noted by the reviewer. We speculate that this compartmentalization may be due to varying degrees of coagulation in different regions of the blood meal. However, this remains a speculative observation without experimental evidence, and thus, we prefer not to discuss this phenomenon further to avoid confusion.

Moreover, it's important to note that the staining protocol used in this assay is not specific for peritrophic matrix. We provide two examples below, showing midguts before and after supplementation with AgApyrase, where compartmentalization of the fibrin signal is evident. Insets of higher magnification highlight areas where we predict the peritrophic matrix to be located (indicated by red arrows). At this stage, we lack evidence to suggest that Apyrase contributes to peritrophic matrix degradation, a phenomenon that warrants investigation in future studies.

13. Lines 227-229 and 250-260: Could apyrase act on platelets within the mouse before blood ingestion by the mosquito? Consider conducting blood feeding (either on mouse or human blood) using SMFAs spiked with the apyrase.

Previous research indicates that apyrase does not directly act on platelets but rather acts on ATP or ADP, and now on tPA. If anything, apyrase is expected to prevent platelet activation and aggregation within the mouse, which should facilitate the

acquisition of platelets during blood feeding. The experiment proposed by the reviewer is emphasized as a central issue discussed in the manuscript, highlighting the limitations of using SMFAs. In these assays, coagulation factors are either absent (when using serum) or are present (when using plasma or whole blood) but are inhibited by anticoagulants. This would create an artificial scenario, complicating the interpretation of the results.

14. In line 275, "both" should replace "either/or."

We have removed either/or and modified the sentence to reflect "both".

15. The argument in lines 303-305 regarding sporozoites regulating apyrase expression to increase the frequency of mosquito bites needs clarification. It would be more intuitive if increased apyrase amounts correlated with successful blood feeding rather than more frequent bites, aligning better with the results on sporozoite infectivity in immunized mice via sporozoite blockade in the skin.

The reduction in apyrase activity observed in previous studies can be attributed to a general decrease in salivary proteins within infected salivary glands, as indicated by our analysis of the saliva proteome from infected and uninfected mosquitoes (refer to the unpublished data figure included in the answer to Reviewer 2. Although we observed a slight decrease in apyrase levels in infected glands, this decrease was not statistically significant and may not be biologically relevant. Given that apyrase is the most abundant salivary protein, this reduction might not have significant biological implications. Furthermore, since the increase in probing behavior by infected mosquitoes might not be directly linked to apyrase activity, we have decided to remove this statement from the manuscript. Additionally, we have eliminated discussion regarding apyrase activity, as further research is required to confirm whether this reduction has any impact on mosquito biting behavior and malaria transmission.

REVIEWERS' COMMENTS

Reviewer #1 (Remarks to the Author):

All my concerns have been addressed.

Please change the word "drive" in the title to "facilitate" or other similar word, as suggested by the 2nd reviewer.

Reviewer #2 (Remarks to the Author):

The authors have satisfactorily addressed my concerns with the initial submission.

We thank you and the reviewers for their valuable feedback towards our manuscript: Mosquito salivary apyrase regulates blood meal hemostasis and drives malaria parasite transmission. Please see below the addressed comments:

REVIEWER COMMENTS

Reviewer #1 (Remarks to the Author):

Please change the word "drive" in the title to “facilitate” or other similar word, as suggested by the 2nd reviewer.

We have edited the manuscript title as suggested by the reviewer.